# Benchmarking Algorithms for Federated Domain Generalization

**Ruqi Bai, Saurabh Bagchi & David I. Inouye**
Elmore Family School of Electrical and Computer Engineering
Purdue University
West Lafayette, IN 47907, USA
`{bai116,sbagchi,dinouye}@purdue.edu`

## Abstract

While prior federated learning (FL) methods mainly consider client heterogeneity, we focus on the *Federated Domain Generalization (DG)* task, which introduces train-test heterogeneity in the FL context. Existing evaluations in this field are limited in terms of the scale of the clients and dataset diversity. Thus, we propose a Federated DG benchmark that aim to test the limits of current methods with high client heterogeneity, large numbers of clients, and diverse datasets. Towards this objective, we introduce a novel data partition method that allows us to distribute any domain dataset among few or many clients while controlling client heterogeneity. We then introduce and apply our methodology to evaluate 14 DG methods, which include centralized DG methods adapted to the FL context, FL methods that handle client heterogeneity, and methods designed specifically for Federated DG on 7 datasets. Our results suggest that, despite some progress, significant performance gaps remain in Federated DG, especially when evaluating with a large number of clients, high client heterogeneity, or more realistic datasets. Furthermore, our extendable benchmark code will be publicly released to aid in benchmarking future Federated DG approaches.

## 1 Introduction

*Domain generalization* (DG) (Blanchard et al., 2011) formalizes a special case of *train-test heterogeneity* in which the training algorithm has access to data from multiple source domains but the ultimate goal is to perform well on test data coming from a different distribution from training distribution—i.e., a type of out-of-distribution generalization instead of the standard in-distribution generalization. While most prior DG work focuses on centralized algorithms, another natural context is federated learning (FL) (Konečnỳ et al., 2016), which is a distributed machine learning context that assumes each client or device owns a local dataset. These local datasets could exhibit heterogeneity, which we call *client heterogeneity* (e.g., class imbalance between clients). Although train-test heterogeneity (in DG) and client heterogeneity (in FL) are independent concepts, both could be naturally defined in terms of domain datasets. For example, suppose a network of hospitals aimed to use FL to train a model to predict a disease from medical images. Because the equipment is different across hospitals, it is natural to assume that each hospital contains data from different domains or environments (or possibly a mixture of domains if it is a large hospital)—this is a case of domain-based client heterogeneity. Yet, the trained model should be robust to changes in equipment within a hospital or to deployment in a new hospital that joins the network—both are cases of domain-based train-test heterogeneity. The interaction between these types of heterogeneity produces new algorithmic and theoretic challenges yet it may also produce new insights and capabilities. Solutions to Federated DG with both types of heterogeneity could increase the robustness and usefulness of FL approaches because the assumptions more naturally align with real-world scenarios rather than assuming the datasets are i.i.d. This could enable training on partial datasets, increase robustness of models to benign spatial and temporal shifts, and reduce the need for retraining.

In the centralized regime, various approaches have been proposed for DG, including feature selection, feature augmentation, etc. Most of these methods are not applicable in the FL regime which poses unique challenges. In the FL regime, client heterogeneity has long been considered

a statistical challenge since FedAvg (McMahan et al., 2017), where it experimentally shows that FedAvg effectively mitigate some client heterogeneity. There are many other extensions based on the FedAvg framework tackling the heterogeneity among clients in FL, for example using variance reduction method (Karimireddy et al., 2020). An alternative setup in FL, known as the personalized setting, aims to learn personalized models for different clients to tackle heterogeneity, for example Hanzely et al. (2020). There are also unsupervised FL methods tackling domain translation instead of classification (Zhou et al., 2022; Wang et al., 2023). However, none of these works consider model robustness under domain shift between training and testing data. Recently, a few works in the FL regime tackling DG (Liu et al., 2021; Zhang et al., 2021; Nguyen et al., 2022; Tenison et al., 2022b) have been proposed, however their evaluations are limited in the following senses: **1)** The evaluation datasets are limited in the number and diversity of domains. **2)** The evaluations are restricted to the case when the number of clients is equal to the number of domains, which may be an unrealistic assumption (e.g., a hospital that has multiple imaging centers or a device that is used in multiple locations). The case when clients number might be massive are of both theoretical and application interests. **3)** None of the works consider the influence of the effect of the number of communication rounds. We provide an overview of the tasks in Table 1, considering both the heterogeneity between training and testing datasets (standard vs. domain generalization) and among clients (domain client heterogeneity). While some studies have addressed the standard supervised learning task, there is a need for a fair evaluation to understand the behavior of domain generalization algorithms in the FL context under those new challenges.

There are several benchmark datasets available for evaluating domain generalization (DG) methods in the centralized setting. These benchmarks, such as DomainBed (Gulrajani and Lopez-Paz, 2020) and WILDS (Koh et al., 2021), provide multiple datasets that are suitable for assessing the performance of DG algorithms. However, they did not explicitly consider the unique challenges that arise in the federated learning (FL) setting. On the other hand, there are also benchmarks specifically designed for FL. For instance, the LEAF (Caldas et al., 2018) and FLamby (Terrail et al., 2022) benchmark provides a standardized framework for evaluating FL algorithms. They include several datasets from various domains and allows researchers to assess the performance of their algorithms in a realistic FL scenario. Another benchmark for FL is PFLBench (Chen et al., 2022), which focuses on evaluating personalized FL methods. PFLBench provides 12 datasets containing various applications. Though these FL-based benchmarks consider statistical heterogeneity, they fail to consider the DG task adequately. Moreover, the level of statistical heterogeneity present in these datasets is insufficient for proper DG evaluation. In summary, DG benchmarks do not consider FL challenges, and FL benchmarks do not consider DG challenges.

**Major contributions:** We develop a benchmark methodology for evaluating Federated DG with various client heterogeneity contexts and diverse datasets, and we evaluate representative Federated DG approaches with this methodology. **1)** We develop a novel method to partition dataset across any number of clients that is able to control the heterogeneity among clients. (see Section 3). **2)** We propose the first Federated DG benchmark methodology including four important dimensions of the experimental setting (see Section 4). **3)** We compare three broad approaches to Federated DG: centralized DG methods naïvely adapted to FL setting, FL methods developed for client heterogeneity (e.g., class imbalance), and recent methods specifically designed for Federated DG on 7 diverse datasets. Our results indicate that there still exist significant gaps and open research directions in Federated DG. **4)** We release an extendable open-source library for evaluating Federated DG methods.

**Notation:** Let $[A] := \{1, 2, \cdots, A\}$ denote the set of integers from 1 to $A$. Let $d \in [D]$ denote the $d$-th domain out of $D$ total training domains and let $c \in [C]$ denote the $c$-th client out of $C$ total clients. Let $\mathcal{D} \subseteq [D]$ denote a subset of domain indices. Let $\mathcal{L}(\theta; p)$ denote a generic objective with parameters $\theta$ given a distribution $p$, which is approximated via samples from $p$. Let $p_d$ and $p_c$ denote the distribution of the $d$-th domain and the $c$-th client, respectively. Let $\mathcal{S}$ denote a set of samples.

## 2 BACKGROUND: FEDERATED DOMAIN GENERALIZATION METHODS

In this section, we briefly introduce the problem backup and setup. Furthermore, evaluation on FL regimes often requires data partition methods to distribute training data to each client. Thus we also introduce the existing partition methods in this section.

## 2.1 PROBLEM BACKGROUND AND SETUP

**Domain generalization:** Unlike standard ML which assumes the training and test data are independent and identically distributed (i.i.d.), the ultimate goal of DG is to minimize the average or worst-case loss of test domain distributions using only samples from the training domain distributions. Formally, given a set of training domain distributions $\{p_d : d \in \mathcal{D}_{\text{train}}\}$, minimize the average or worst case loss over test domain distributions, i.e.,

$$\min_\theta \frac{1}{|\mathcal{D}_{\text{test}}|} \sum_{d \in \mathcal{D}_{\text{test}}} \mathcal{L}(\theta; p_d) \quad \text{or} \quad \min_\theta \max_{d \in \mathcal{D}_{\text{test}}} \mathcal{L}(\theta; p_d), \tag{1}$$

where $\mathcal{L}(\theta; p_d) = \mathbb{E}_{(x,y) \sim p_d}[\ell(x, y; \theta)]$ where $\ell$ is a per-sample loss function such as squared or cross-entropy loss. Those two objectives are the ultimate goal for domain generalization. In the real-world setting, we never have access to $\mathcal{D}_{\text{test}}$. There are works establishing objectives to approximate those two objectives. For example, Fish (Shi et al., 2021) is constructed using average training loss with a penalty, and IRM (Arjovsky et al., 2019), GroupDRO (Sagawa et al., 2019) are inspired from the worst-case loss over a set of data distributions constructed from the training domains.

There are mainly two kinds of DG test scenarios. The first scenario is the general DG where the test dataset form a new domain. Another scenario is called subpopulation shift where the test set is a subpopulation of the training distributions with the goal of better performance on the worst-case domain[1] DG is challenging where it breaks the i.i.d. assumptions in traditional machine learning.

**Federated DG:** *Federated* DG represents an intuitive progression from centralized DG. Distributed learning is pivotal in numerous real-world applications, and its inherent architecture often results in multi-domain data. The quest for improved generalization is instinctive in this context. However, *Federated* DG introduces add two layers of complexity. First, the heterogeneous client distribution are harder to converge even without considering the train-test heterogeneity (Zhao et al., 2018; McMahan et al., 2017; Karimireddy et al., 2020; Yu et al., 2019; Basu et al., 2019; Wang et al., 2019; Li et al., 2019). Second, privacy considerations and communication constraints typically prohibit the direct comparison of data across domains, thereby limiting the applicability of many conventional centralized DG methods.

Formally, the FL problem can be abstracted as follows:

$$\forall c \in [C], \underbrace{\theta_c = \texttt{Opt}(\mathcal{L}(\theta; p_c), \theta_{\text{init}} = \theta_{\text{global}})}_{\text{Locally optimize given local distribution } p_c} \text{ and } \underbrace{\theta_{\text{global}} = \texttt{Agg}(\theta_1, \theta_2, \cdots, \theta_C)}_{\text{Aggregate client model parameters on server}},$$

where the client distributions may be homogeneous (i.e., $\forall (c, c'), p_c = p_{c'}$) or heterogeneous (i.e., $\exists c \neq c', p_c \neq p_{c'}$), $\texttt{Opt}$ minimizes an objective $\mathcal{L}(\theta; p_c)$ initialized at $\theta_{\text{init}}$. The most common objective in Federated learning is empirical risk minimization (ERM), which minimizing the average loss over the given dataset. $\texttt{Agg}$ aggregates the client model parameters, where the most common aggregator is simply a (weighted) average of the client parameters, which corresponds to FedAvg (McMahan et al., 2017).

**Domain-based client heterogeneity:** While previous client heterogeneity (i.e., $\exists c \neq c', p_c \neq p_{c'}$) is often expressed as label imbalance, i.e., $p_c(y) \neq p_{c'}(y)$, we make a *domain-based client heterogeneity* assumption that each client distribution is a (different) mixture of train domain distributions, i.e., $p_c(x, y) = \sum_{d \in \mathcal{D}_{\text{train}}} w_{c,d} \, p_d(x, y)$ where $w_{c,d}$ is the weight of the $d$-th domain for the $c$-th client. At one extreme, FL with i.i.d. data would be equivalent to the mixture proportions being the same across all clients, i.e., $\forall c, c', d, w_{c,d} = w_{c',d}$, which we call the *homogeneous* setting. On the other extreme where the heterogeneity is maximum given the client number, we call *complete heterogeneity* (in Section 3. Table 1 summarizes the train-test heterogeneity in DG and the domain-based client heterogeneity from the FL context, where we focus on Federated DG.

## 2.2 CURRENT DATA PARTITION METHODS

In the FL context, the method for partitioning training data is crucial. As of now, there are three primary techniques for dataset partitioning. These methods are all fall in short in some properties we need. See Section 3 for a comprehensive analysis. In this section, we primarily present these methods as a background. Shards partitioning (McMahan et al., 2017) is a deterministic partition

---

[1]In Wilds (Koh et al., 2021), they treat subpopulation shift as another kind of train-test heterogeneity.

Table 1: Domain-based Federated DG considers the *domain heterogeneity* both among the clients' local datasets (rows) and between the training and test datasets (columns). This paper focuses on the DG setting (right column).

| | | Between train and test datasets (train-test heterogeneity) | |
|---|---|---|---|
| | | Standard supervised learning | **Domain generalization (our focus)** |
| Among clients | Homogeneous ($\lambda = 1$) | FL with homogeneous clients | Federated DG with homogeneous clients |
| | Heterogeneous ($0 < \lambda < 1$) | FL with heterogeneous clients | Federated DG with heterogeneous clients |
| | Complete Heterogeneity ($\lambda = 0$) | FL with complete heterogeneity | Federated DG with complete heterogeneity |

method. The datasets are initially ordered according to their labels. Then the datasets are partitioned into $2 \times C$ shards, and each client $c \in [C]$ is allocated 2 shards. Dirichlet Partitioning (Yurochkin et al., 2019; Hsu et al., 2019) is a stochastic partition method where each client hold the same number of samples but the proportion among labels are stochastic following a probability vector sampled from the Dirichlet distribution $\text{Dir}(\alpha p)$, where $p$ represents the prior label distribution on the whole training dataset, and $\alpha$ control the heterogeneity level. As $\alpha \to 0$, each client predominantly holds samples of a single label. Conversely, as $\alpha \to \infty$, the distribution of labels for each client becomes identical. Semantic Partitioning (Yuan et al., 2022) tries to create client heterogeneity on the features of data samples. Firstly, the data is processed through a pretrained model. The outputs from this model are then used to fit a Gaussian Mixture Model that features $K$ clusters for each class $Y$. These $KY$ clusters are then merged iteratively using an optimal bipartite for random label pairs.

## 3    HETEROGENEOUS PARTITIONING METHOD

In this section, to thoroughly assess current Federated DG methods, we formulate an optimization problem that an ideal partition method should aim to solve. We then demonstrate that previous partition methods (subsection 2.2) are not feasible solutions. Subsequently, we introduce a new partition method, termed Heterogeneous Partitioning, which is an approximate optimal solution. Generally, our partition method effectively handles partitioning $D$ types of integer-numbered objects into $C$ groups. It's broadly applicable, suitable for domain adaptation, ensuring fairness in Federated Learning (FL), and managing non-iid FL regimes.

Assume we have a set of all samples $\mathcal{S} = \{(x_i, d_i)\}_{i=1}^n$ and let $\mathcal{S}_c$ denote the set of samples assigned to the $c$-th client. For each client $c \in [C]$, define its domains as the set of domains from which $c$ has at least one sample., i.e., $\mathcal{D}_c = \{d \in [D] : \exists (x_i, d_i) \in \mathcal{S}_c, d_i = d\}$. Denote $\mathcal{P}$ as a partition method.

Table 2: Comparison of different partition methods.

| | $C_1$ | $C_2$ |
|---|---|---|
| Shards | No | Yes |
| Dirichlet | Yes | No |
| Semantic | No | Yes |
| Ours | Yes | Yes |

For Federated DG, it is common that clients have datasets collected from disjoint domains Koh et al. (2021). Therefore, it's crucial that $\mathcal{P}$ can generate the strongest possible heterogeneity given the datasets and clients to reflect those common scenarios. This requirement can be encoded as either a single domain per client or no intersection of client domains, depending on the numbers of clients and domains (see constraint $C_1$). Furthermore, in distributed setting, to ensure a reasonable comparison with respect to domain generalization in the centralized setting, the total datasets received over the distributed system should match that of the centralized setting; and due to the privacy protocol, clients should not share the datasets. Thus, we need to preclude the partitions $\mathcal{P}$ where some data from $\mathcal{S}$ is not used by any client or clients either have duplicated data, which translate to constraint $C_2$. Thus, the feasible region of partition $\mathcal{P}$ is defined as $\mathcal{P} \in C_1 \cap C_2$.

Complete Heterogeneity (constraint $C_1$): $\mathcal{P}$ can generate the possibly strongest domain heterogeneity given the datasets and clients, i.e., $\mathcal{P} \in C_1$ where $C_1$ is defined as

$$C_1 \triangleq \left\{ \mathcal{P} \,\middle|\, \text{if } C > D, \text{ then } |\mathcal{D}_c| = 1, \forall c\,;\ \text{if } C \leq D, \text{ then } |\mathcal{D}_c \cap \mathcal{D}_{c'}| = 0, \forall c \neq c'. \right\} \quad (2)$$

True Partition (constraint $C_2$): $\mathcal{P}$ must produce true set partitions $\{\mathcal{S}_c\}, c \in [C]$. A true set partition is a non-empty, exhaustive, and pairwise disjoint family of sets, i.e., $\mathcal{P} \in C_2$ where $C_2$ is defined as

$$C_2 \triangleq \left\{ \mathcal{P} \;\middle|\; \mathcal{S}_c \neq \emptyset, \forall c, \bigcup_{c=1}^{C} \mathcal{S}_c = \mathcal{S}, \text{ and } \mathcal{S}_c \cap \mathcal{S}_{c'} = \emptyset, \forall c \neq c'. \right\}. \tag{3}$$

The primary goal of this benchmark is to investigate the impact of client domain heterogeneity on the performance of various Federated DG methods. When considering client heterogeneity in a federated setting, a trivial example would be one client holding 99% of the total data. This scenario essentially mirrors centralized DG, which is already well-understood. However, the genuine uncertainty arises when there's heightened client heterogeneity introduced into domain generalization, especially when data sizes across different clients are nearly equal. In such a situation, the influence of client heterogeneity is at its peak. Our aim is to reduce the sample size imbalance between clients, which we encode as the empirical variance among them. Given this, we define an ideal partition $\widehat{\mathcal{P}}$ as the solution for the following problem:

$$\widehat{\mathcal{P}} \in \operatorname*{arg\,min}_{\mathcal{P} \in C_1 \cap C_2} \frac{1}{C-1} \sum_{c=1}^{C} \|n_c - \bar{n}\|_2^2, \tag{4}$$

where $n_c = |\mathcal{S}_c|$, $\bar{n} = \frac{1}{C} \sum_c n_c$. Notice that in many scenarios, it is not always able to achieve 0 loss, because $n_c \equiv \bar{n}$ may not be feasible for constraints 1 and 2. For example, consider the case of complete heterogeneity with $C = D$ but imbalance of domains, it is impossible to achieve balance $n_c \equiv \bar{n}$ and complete hetereogeneity $|\mathcal{D}_c| = 1$. Most previous partitions forces $n_c \equiv \bar{n}$ thus violating the constraints. In particular, Dirichlet Partitioning violates $C_2$. Shard and semantic violate $C_1$. We compare different partition methods in Table 2. See subsection A.2 for a detailed discussion on them.

We construct Heterogeneous Partitioning $\{\mathcal{P}_\lambda\}, \lambda \in [0, 1]$, to solve Eqn. 4. It is parameterized by $\lambda$, where it can generate Complete Hetereogeneity ($\lambda = 0$) and always satisfy True Partition (for all $\lambda \in [0, 1]$) by construction; further it aims to achieve the best possible balance given these constraints. Heterogeneous Partitioning allows samples allocation from multiple domains across an arbitrary number of clients $C$ while controlling the amount of domain-based client heterogeneity via parameter $\lambda$, ranging from homogeneity to complete heterogeneity. In addition, when $\lambda = 0$, Heterogeneous Partitioning is optimal for Eqn. 4 when there's more clients than domains $C > D$, Further, when $C \leq D$, Eqn. 4 is NP-hard problem proposed by Graham (1969) and Heterogeneous Partitioning is a linear time greedy approximation. The proof is deferred to subsection A.1. Heterogeneous Partitioning contains the following two steps, see Algorithm 1 for pseudo code.

**Step 1: Constructing complete heterogeneity.** We explicitly construct $\mathcal{P}_0$ to satisfy $C_1$ while greedily trying to balance the variance. The key idea to create the complete heterogeneity is to partition one domain among the fewest possible clients.

We adopt the following approaches depending on the size of the network $C$ and total domains $D$.

*Case I*: If $C \leq D$, the domains are sorted in a descending order according to number of sample size in each domain, and are iteratively assigned to the client $c^*$ which currently has the smallest number of training samples, that is

$$c^* \triangleq \operatorname*{arg\,min}_{c \in [C]} \sum_{d' \in \mathcal{D}_c} n_{d'}, \tag{5}$$

where $n_d$ denotes the total sample size of domain $d$. In this case, $\mathcal{P}_0$ ensures that no client shares domains with the others, i.e., $|D_c \cap D'_c| = 0$, but attempts to balance the number of training samples between clients.

*Case II*: If $C > D$, we first assign all the domains one by one to the first $D$ clients; then, starting from client $c = D + 1$, we divide the currently largest average domain $d^*$ between the new client and the original clients associated with it. That is, for all $c \in \{D + 1, D + 2, \ldots, C\}$, we assign $d^*$ to $c$, and reassign the same sample sizes for those $c' \in \{1, \ldots, c\}$ which share the domain $d^*$ as

$$d^* \in \operatorname*{arg\,max}_{d \in [D]} \frac{n_d}{\sum_{c'=1}^{C} \mathbb{1}[d \in \mathcal{D}_{c'}]}, \quad \frac{n_{d^*}}{\sum_{c'=1}^{C} \mathbb{1}[d^* \in \mathcal{D}_{c'}]}, \tag{6}$$

where rounding to integers is carefully handled when not perfectly divisible. Notice that in this case, some clients may share one domain, but no client holds two domains simultaneously, that is

$|D_c| = 1$, for $c \in [C]$. In this case, $\mathcal{P}_0$ attempts to balance the number of samples across clients as much as possible while partitioning one domain among the fewest possible clients.

**Step 2: Computing domain sample counts for each domain with interpolation.** We define $\mathcal{P}_\lambda$ through the sample counts for each client $c \in [C]$ per domain $d \in [D]$, denoted as $n_{d,c}(\lambda)$ based on the balancing parameter $\lambda \in [0, 1]$ :

$$\mathcal{P}_\lambda: \quad n_{d,c}(\lambda) = \underbrace{\lambda \frac{n_d}{C}}_{\text{homogeneous}} + \underbrace{(1 - \lambda)\frac{\mathbb{1}[d \in \mathcal{D}_c]}{\sum_{c'=1}^{C} \mathbb{1}[d \in \mathcal{D}_{c'}]} n_d}_{\text{complete heterogeneous}}, \tag{7}$$

where $\mathcal{D}_c, c \in [C]$ are constructed by $\mathcal{P}_0$ Step 1. This is simply a convex combination between homogeneous clients ($\lambda = 1$) and extreme heterogeneous ($\lambda = 0$). A smaller value of $\lambda$ corresponds to a more heterogeneous case. Given the number of samples per client $i \in [C]$ per domain $d \in [D]$, we simply sample $n_{d,c}$ without replacement from the corresponding domain datasets and build up the client datasets. Step 2 ensures our partition $\mathcal{P}_\lambda$ satisfy the constraint $C_2$, because all samples are selected (as proven by $\sum_c n_{d,c} = n_d$) and there is no overlap (sample without replacement), and no client has zero samples given $|D_c| > 0$ in the $\lambda = 0, 1$ cases.

## 4 BENCHMARK METHODOLOGY AND EVALUATIONS

In this section, we aim to conduct a comprehensive evaluation of the Federated DG task by considering four distinct dimensions of the problem setup equipped by Heterogeneous Partitioning. The four dimensions are **1)** dataset type; **2)** data difficulty; **3)** domain-based client heterogeneity; and **4)** the number of clients. This section is organized as follows: we introduce our benchmark datasets subsection 4.1 covering the first two dimensions, next we introduce 14 methods we included in our evaluation from three different approaches subsection 4.2, then we introduce our benchmark setting and the evaluation results on all selected methods in subsection 4.3 brought the domain-based client heterogeneity and number of clients into consideration.

### 4.1 DATASET TYPE AND DATASET DIFFICULTY METRICS

While most Federated DG work focuses on standard image-based datasets, we evaluate methods over a diverse selection of datasets. These datasets encompass multi-domain datasets from simpler, pseudo-realistic ones to the considerably more challenging realistic ones. Specifically, our study includes five image datasets and two text datasets. Additionally, within these 7 datasets, we include one subpopulation shift within image datasets (CelebA) and another within text datasets (Civilcomments). Furthermore, our dataset selections span a diverse range of subjects including general objects, wild camera traps, medical images, human faces, online comments, and programming codes. We also introduce dataset difficulty metrics to measure the empirical challenges of each dataset in the Federated DG task.

**Dataset Difficulty Metrics.** To ensure a comprehensive evaluation of current methods across different difficulty levels, we have curated a range of datasets with varying complexities. We define two dataset metrics to measure the dataset difficulty with respect to the DG task and with respect to the FL context using the baseline objective ERM. For DG difficulty, we compute $R_{\text{DG}}$, the ratio of the ERM performance without and with samples from the test domain (i.e., the later is able to "cheat" by seeing part of test domain samples during training). For FL difficulty, we attempt to isolate the FL effect by computing $R_{\text{FL}}(\lambda)$, the ratio of ERM-based FedAvg $\lambda$ client heterogeneity over centralized ERM on *in-domain* test samples. These dataset difficulty metrics can be formalized as follows, for all $\lambda \in [0, 1]$

$$R_{\text{DG}} \triangleq \frac{\texttt{ERM\_Perf}(\mathcal{S}_{\text{DG-train}}, \mathcal{S}'_{\text{DG-test}})}{\texttt{ERM\_Perf}(\mathcal{S}_{\text{DG-train}} \cup \mathcal{S}''_{\text{DG-test}}, \mathcal{S}'_{\text{DG-test}})}, R_{\text{FL}}(\lambda) \triangleq \frac{\texttt{FedAvg\_Perf}(\mathcal{S}_{\text{DG-train}}, \mathcal{S}_{\text{IN-test}}; \lambda)}{\texttt{ERM\_Perf}(\mathcal{S}_{\text{DG-train}}, \mathcal{S}_{\text{IN-test}})},$$

where $\texttt{ERM\_Perf}$ is the performance of ERM using the first argument as training and the second for test, $\texttt{FedAvg\_Perf}$ is similar but with the client heterogeneity parameter $\lambda$, $\mathcal{S}_{\text{DG-train}}$ denotes samples from the training domains $\mathcal{D}_{\text{train}}$, $\mathcal{S}_{\text{DG-test}}$ denotes samples from the test domains $\mathcal{D}_{\text{test}}$, and $\mathcal{S}'_{\text{DG-test}}$ and $\mathcal{S}''_{\text{DG-test}}$ are $20\%, 80\%$ partition respectively of $\mathcal{S}_{\text{DG-test}}$. For $R_{\text{FL}}(\lambda)$, we use $\mathcal{S}_{\text{IN-test}}$ (test samples from the training domains) instead of $\mathcal{S}_{\text{DG-test}}$ to isolate the FL effect from the DG effect. We

apply these metrics in our 7 selected datasets and include the summary table in Table 6. Smaller $R_{\text{DG}}$ and $R_{\text{FL}}(\lambda)$ indicate a more challenging dataset. For instance, FEMNIST has $R_{\text{DG}} = 1$ indicates the lack of domain heterogeneity. To contrast, IwildCam and CivilComments have small $R_{\text{FL}}$ showing the challenges from the FL side.

## 4.2 BENCHMARK METHODS

In this benchmark study, we explore three categories of Federated DG methods. The first category is centralized DG methods adapted into FL regimes. The second category is FL methods tackling client heterogeneity and the third category is Federated DG methods. Please see Appendix B for a detailed discussion on current available methods to solve Federated DG. To provide a comprehensive evaluation, we assess the performance of several representative methods from each of these categories. specifically, we choose IRM (Arjovsky et al., 2019), Fish (Shi et al., 2021), Mixup (Zhang et al., 2017), MMD (Gretton et al., 2006), Coral (Sun and Saenko, 2016), GroupDRO (Sagawa et al., 2019) and adapted them into FL regime by applying them in every local clients and keeping the aggregation identical to FedAvg, i.e., weighted averaging aggregation. All those methods collapse to FedAdam if locally there is only one domain available. We consider 4 methods. FedDG (Liu et al., 2021), FedADG (Zhang et al., 2021), FedSR (Nguyen et al., 2022) and FedGMA (Tenison et al., 2022a), which are naturally designed to solve Federated DG. We also consider FedProx (Li et al., 2020) and Scaffold (Karimireddy et al., 2020) from the Federated methods tackling client heterogeneity approach. All those methods will be compared to two baselines. ERM objective with Adam optimizer (Kingma and Ba, 2014) and its FL counterpart FedAdam (Reddi et al., 2020), which is a variant of FedAvg (McMahan et al., 2017).

## 4.3 MAIN RESULTS

In this section, we present the performance results of $14$ representative methods, derived from distinct research areas, on 7 diverse datasets. For each dataset, we fix the total computation and communication rounds for different methods for a fair comparison.

**Client number** Equipped with our Heterogeneous Partitioning, we can explore various levels of client heterogeneity and relax the assumption that $C = D$ so that we can leverage both pseudo-realistic and real-world datasets and evaluate methods on larger scale of FL clients. In particular, we set the number of clients to 100.

**Validation domain** In DG task, we cannot access the test domain data. However, we are particularly concerned about the model performance outside the training domains, thus we preserve a small portion of the domains we can access as held-out validation domains, and the held-out validation domains are used for model selection and early stopping. Please see Appendix D for more detail.

After training, we choose the model according to the early-stopping at the communication round which achieves the best held-out-domain performance, and finally we evaluate the performance on the test-domain in Table 3, Table 4 and Table 5. See subsection D.4 for detailed hyperparameters choices. We make the following remarks on the main results from Table 3, Table 4 and Table 5. Because FEMNIST has low DG difficulty, we defer the results on FEMNIST in the Appendix (Table 11).

*Remark* 4.1. **FedAvg with an ERM objective is a strong baseline, especially on the general DG datasets.** We observe FedAvg-ERM outperforms other methods multiple times. It still servers a strong baseline that is challenging to beat across general DG datasets, similar to the centralized case stated in DomainBed (Gulrajani and Lopez-Paz, 2020) and WILDS (Koh et al., 2021). We recommend always including FedAvg as a baseline in all future evaluations.

*Remark* 4.2. **Most centralized DG methods degrade in the FL setting.** For image datasets, the DG methods adapted to the FL setting (except for GroupDRO) show significant degradation in performance compared to their centralized counterparts as can be seen when comparing the $C = 1$ column to the $C > 1$ columns in Table 3. Further, degradation can be seen in PACS and CelebA when moving from the homogeneous client setting ($\lambda = 1$) to the heterogeneous client setting ($\lambda = 0.1$).

*Remark* 4.3. **FL methods tackling client heterogeneity help convergence.** Notably, IWildCam and CivilComments datasets bring greater challenges in the model convergence as Table 6. In this scenario, FL methods tackling the client heterogeneity are able to better tackle this challenge while other methods all failed in this context, see Table 3 and Table 5. Given the fact that this is a

Table 3: Test accuracy on PACS and IWildCam dataset with held-out-domain validation where FedAvg-ERM is the simple baseline (B). "-" means the method is not applicable in that context. Bold is for best and italics is for second best in each column. We report the standard deviation among 3 runs. Please see the Table 8 in the appendix for higher precision report.

| | | PACS ($D = 2$) | | | | IWildCam ($D = 243$) | | | |
|---|---|---|---|---|---|---|---|---|---|
| | | $C = 1$ | $C = 100$ (FL) | | | $C = 1$ | $C = 100$ (FL) | | |
| | | (centralized) | $\lambda = 1$ | $\lambda = 0.1$ | $\lambda = 0$ | (centralized) | $\lambda = 1$ | $\lambda = 0.1$ | $\lambda = 0$ |
| B | FedAvg-ERM | $\mathbf{0.94 \pm 0.01}$ | $\mathbf{0.95 \pm 0.02}$ | $\mathbf{0.96 \pm 0.02}$ | $\mathbf{0.95 \pm 0.02}$ | $0.34 \pm 0.00$ | $\mathbf{0.30 \pm 0.00}$ | $0.25 \pm 0.00$ | $\mathbf{0.20 \pm 0.01}$ |
| DG Adapted | IRM | $0.92 \pm 0.01$ | $0.92 \pm 0.04$ | $0.86 \pm 0.04$ | - | $0.32 \pm 0.00$ | $0.20 \pm 0.01$ | $0.18 \pm 0.01$ | - |
| | Fish | $0.94 \pm 0.02$ | $0.65 \pm 0.13$ | $0.35 \pm 0.11$ | - | $\mathbf{0.35 \pm 0.00}$ | $0.18 \pm 0.00$ | $0.15 \pm 0.01$ | - |
| | Mixup | $0.92 \pm 0.01$ | $0.89 \pm 0.05$ | $0.83 \pm 0.03$ | - | $0.33 \pm 0.01$ | $0.09 \pm 0.01$ | $0.07 \pm 0.01$ | - |
| | MMD | $0.93 \pm 0.02$ | $0.92 \pm 0.04$ | $0.86 \pm 0.04$ | - | $0.32 \pm 0.00$ | $0.19 \pm 0.01$ | $0.16 \pm 0.01$ | - |
| | DeepCoral | $0.93 \pm 0.01$ | $0.92 \pm 0.04$ | $0.86 \pm 0.04$ | - | $0.33 \pm 0.01$ | $0.19 \pm 0.01$ | $0.16 \pm 0.01$ | - |
| | GroupDRO | $0.93 \pm 0.01$ | $0.94 \pm 0.03$ | $0.95 \pm 0.02$ | - | $0.21 \pm 0.00$ | $0.13 \pm 0.01$ | $0.20 \pm 0.01$ | - |
| FL | FedProx | - | $0.90 \pm 0.03$ | $0.89 \pm 0.03$ | $0.90 \pm 0.02$ | - | $0.25 \pm 0.00$ | $0.20 \pm 0.00$ | $0.17 \pm 0.00$ |
| | Scaffold | - | $0.90 \pm 0.03$ | $0.89 \pm 0.02$ | $0.90 \pm 0.02$ | - | $0.28 \pm 0.00$ | $\mathbf{0.26 \pm 0.00}$ | $0.18 \pm 0.01$ |
| | AFL | - | $0.93 \pm 0.03$ | $0.92 \pm 0.04$ | $0.91 \pm 0.04$ | - | $0.26 \pm 0.01$ | $0.16 \pm 0.01$ | $0.03 \pm 0.00$ |
| FDG | FedDG | - | $0.93 \pm 0.02$ | $0.95 \pm 0.02$ | $0.94 \pm 0.02$ | - | $0.27 \pm 0.00$ | $0.24 \pm 0.00$ | $0.17 \pm 0.00$ |
| | FedADG | - | $0.94 \pm 0.01$ | $0.94 \pm 0.00$ | $0.94 \pm 0.01$ | - | $0.26 \pm 0.01$ | $0.25 \pm 0.01$ | $0.16 \pm 0.00$ |
| | FedSR | - | $0.64 \pm 0.16$ | $0.55 \pm 0.09$ | $0.54 \pm 0.09$ | - | $0.18 \pm 0.01$ | $0.14 \pm 0.01$ | $0.09 \pm 0.00$ |
| | FedGMA | - | $0.75 \pm 0.05$ | $0.73 \pm 0.09$ | $0.72 \pm 0.01$ | - | $0.22 \pm 0.00$ | $0.15 \pm 0.00$ | $0.09 \pm 0.00$ |

common challenge in Federated DG, there's a need for future research to simultaneously enhance the performance with both client heterogeneity and train-test heterogeneity.

*Remark* 4.4. **Federated DG methods still in need.** Upon evaluating current Federated DG methods on larger network scales, they all fail to outperform ERM. Among the four methods we evaluating, FedDG performs the best and could achieve higher accuracy on PACS with $\lambda = 0$. However, FedDG requires sharing all local datasets amplitude information to the aggregation server, thus brings privacy concerns. FedADG methods contains tens of hyperparameters, and due to the nature of GAN (Goodfellow et al., 2020), it is challenging to optimize. FedSR and FedGMA also did not outperform ERM objective. It shows a clear demand for further improvement.

*Remark* 4.5. **Addressing the subpopulation shift could be an initial step.** In the centralized setting, all evaluated methods surpassed the performance of ERM on two subpopulation shift datasets: CelebA and CivilComments. However, in a Federated context, only FedDG managed to outpace ERM on CelebA, and yet it couldn't match the performance of centralized approaches. Further exploration in federated DG is in demand to bridge this gap.

*Remark* 4.6. **The performance of real-world data significantly degrades as $\lambda$ decreases.** This can be seen from IWildCam and Py150 dataset at Table 3 and Table 5. While it is challenging and expensive to run models for IWildCam and Py150, they show the largest differences between methods and demonstrates the real-world challenge of Federated DG. We suggest including IWildCam and Py150 in most future DG evaluations given their unique nature across datasets.

Table 4: Test accuracy on CelebA and Camelyon17 dataset with held-out-domain validation where FedAvg-ERM is the simple baseline (B). "-" means the method is not applicable in that context. Bold is for best and italics is for second best in each column. We report the standard deviation among 3 runs. Please see the Table 9 in the appendix for higher precision report.

| | | CelebA ($D = 2$) | | | | Camelyon17 ($D = 3$) | | | |
|---|---|---|---|---|---|---|---|---|---|
| | | $C = 1$ | $C = 100$ (FL) | | | $C = 1$ | $C = 100$ (FL) | | |
| | | (centralized) | $\lambda = 1$ | $\lambda = 0.1$ | $\lambda = 0$ | (centralized) | $\lambda = 1$ | $\lambda = 0.1$ | $\lambda = 0$ |
| B | FedAvg-ERM | $0.77 \pm 0.04$ | $0.61 \pm 0.02$ | $0.52 \pm 0.04$ | $0.45 \pm 0.02$ | $0.90 \pm 0.01$ | $0.95 \pm 0.01$ | $0.95 \pm 0.00$ | $\mathbf{0.95 \pm 0.00}$ |
| DG Adapted | IRM | $\mathbf{0.89 \pm 0.01}$ | $0.71 \pm 0.05$ | $0.74 \pm 0.01$ | - | $0.91 \pm 0.06$ | $0.95 \pm 0.00$ | $0.95 \pm 0.00$ | - |
| | Fish | $0.88 \pm 0.01$ | $0.14 \pm 0.16$ | $0.07 \pm 0.05$ | - | $0.92 \pm 0.01$ | $0.88 \pm 0.01$ | $0.88 \pm 0.01$ | - |
| | Mixup | $0.29 \pm 0.498$ | $0.13 \pm 0.22$ | $0.13 \pm 0.22$ | - | $0.94 \pm 0.01$ | $\mathbf{0.95 \pm 0.00}$ | $0.95 \pm 0.01$ | - |
| | MMD | $0.84 \pm 0.05$ | $0.81 \pm 0.03$ | $0.77 \pm 0.03$ | - | $0.92 \pm 0.03$ | $0.95 \pm 0.00$ | $0.95 \pm 0.01$ | - |
| | DeepCoral | $0.85 \pm 0.04$ | $\mathbf{0.81 \pm 0.02}$ | $\mathbf{0.78 \pm 0.02}$ | - | $\mathbf{0.94 \pm 0.01}$ | $0.95 \pm 0.00$ | $0.95 \pm 0.00$ | - |
| | GroupDRO | $0.87 \pm 0.03$ | $0.81 \pm 0.03$ | $0.76 \pm 0.06$ | - | $0.92 \pm 0.03$ | $0.95 \pm 0.01$ | $\mathbf{0.95 \pm 0.00}$ | - |
| FL | FedProx | - | $0.13 \pm 0.24$ | $0.12 \pm 0.25$ | $0.00 \pm 0.00$ | - | $0.94 \pm 0.00$ | $0.94 \pm 0.00$ | $0.94 \pm 0.01$ |
| | Scaffold | - | $0.73 \pm 0.01$ | $0.78 \pm 0.02$ | $\mathbf{0.80 \pm 0.02}$ | - | $0.94 \pm 0.00$ | $0.93 \pm 0.00$ | $0.93 \pm 0.00$ |
| | AFL | - | $0.81 \pm 0.02$ | $0.83 \pm 0.01$ | $0.83 \pm 0.01$ | - | $0.94 \pm 0.01$ | $0.95 \pm 0.01$ | $0.93 \pm 0.01$ |
| FDG | FedDG | - | $0.56 \pm 0.18$ | $0.53 \pm 0.11$ | $0.46 \pm 0.20$ | - | $0.86 \pm 0.01$ | $0.86 \pm 0.00$ | $0.87 \pm 0.02$ |
| | FedADG | - | $0.67 \pm 0.00$ | $0.67 \pm 0.01$ | $0.60 \pm 0.01$ | - | $0.94 \pm 0.00$ | $0.93 \pm 0.00$ | $0.93 \pm 0.00$ |
| | FedSR | - | $0.38 \pm 0.02$ | $0.36 \pm 0.04$ | $0.38 \pm 0.02$ | - | $0.93 \pm 0.01$ | $0.92 \pm 0.01$ | $0.93 \pm 0.00$ |
| | FedGMA | - | $0.62 \pm 0.01$ | $0.62 \pm 0.02$ | $0.49 \pm 0.01$ | - | $0.90 \pm 0.01$ | $0.85 \pm 0.01$ | $0.82 \pm 0.03$ |

Table 5: Test accuracy on CivilComments and Py150 datasets with held-out-domain validation where FedAvg-ERM is the simple baseline (B). "-" means the method is not applicable in that context. Bold is for best and italics is for second best in each column. FedDG and FedADG are designed for image dataset and thus not applicable for CivilComments and Py150. We report the standard deviation among 3 runs. Please see the Table 10 in the appendix for higher precision report.

| | | CivilComments ($D = 16$) | | | | Py150 ($D = 5477$) | | | |
| | | $C = 1$ | $C = 100$ (FL) | | | $C = 1$ | $C = 100$ (FL) | | |
| | | (centralized) | $\lambda = 1$ | $\lambda = 0.1$ | $\lambda = 0$ | (centralized) | $\lambda = 1$ | $\lambda = 0.1$ | $\lambda = 0$ |
|---|---|---|---|---|---|---|---|---|---|
| B | FedAvg-ERM | $0.54 \pm 0.00$ | $0.36 \pm 0.03$ | $0.35 \pm 0.02$ | $0.33 \pm 0.01$ | $\mathbf{0.68 \pm 0.00}$ | $\mathbf{0.68 \pm 0.00}$ | $\mathit{0.65 \pm 0.00}$ | $\mathit{0.64 \pm 0.00}$ |
| DG Adapted | IRM | $0.64 \pm 0.00$ | $\mathit{0.59 \pm 0.02}$ | $0.53 \pm 0.04$ | - | $\mathit{0.68 \pm 0.00}$ | $\mathit{0.66 \pm 0.00}$ | $\mathbf{0.65 \pm 0.00}$ | $0.63 \pm 0.00$ |
| | Fish | $\mathbf{0.67 \pm 0.00}$ | $0.42 \pm 0.16$ | $0.34 \pm 0.17$ | - | $0.66 \pm 0.00$ | $0.65 \pm 0.00$ | $0.65 \pm 0.00$ | $\mathbf{0.64 \pm 0.00}$ |
| | MMD | $\mathit{0.65 \pm 0.00}$ | $\mathbf{0.63 \pm 0.01}$ | $\mathbf{0.61 \pm 0.01}$ | - | $0.66 \pm 0.00$ | $0.63 \pm 0.00$ | $0.63 \pm 0.00$ | $0.61 \pm 0.00$ |
| | DeepCoral | $0.59 \pm 0.00$ | $0.52 \pm 0.06$ | $0.46 \pm 0.08$ | - | $0.66 \pm 0.00$ | $0.65 \pm 0.00$ | $0.65 \pm 0.00$ | $0.64 \pm 0.00$ |
| | GroupDRO | $0.64 \pm 0.00$ | $0.48 \pm 0.01$ | $\mathit{0.47 \pm 0.00}$ | - | $0.51 \pm 0.00$ | $0.59 \pm 0.00$ | $0.60 \pm 0.00$ | $0.61 \pm 0.00$ |
| FL | FedProx | - | $0.18 \pm 0.03$ | $0.17 \pm 0.03$ | $0.17 \pm 0.04$ | - | $0.64 \pm 0.00$ | $0.63 \pm 0.00$ | $0.61 \pm 0.00$ |
| | Scaffold | - | $0.39 \pm 0.02$ | $0.38 \pm 0.02$ | $\mathit{0.33 \pm 0.01}$ | - | $0.64 \pm 0.00$ | $0.64 \pm 0.00$ | $0.62 \pm 0.00$ |
| | AFL | - | $0.55 \pm 0.02$ | $0.47 \pm 0.01$ | $\mathbf{0.44 \pm 0.02}$ | - | $0.49 \pm 0.00$ | $0.49 \pm 0.00$ | $0.47 \pm 0.00$ |
| FDG | FedSR | - | $0.36 \pm 0.00$ | $0.34 \pm 0.00$ | $0.32 \pm 0.00$ | - | $0.53 \pm 0.00$ | $0.53 \pm 0.00$ | $0.45 \pm 0.00$ |
| | FedGMA | - | $0.21 \pm 0.03$ | $0.20 \pm 0.02$ | $0.20 \pm 0.02$ | - | $0.62 \pm 0.00$ | $0.61 \pm 0.00$ | $0.60 \pm 0.00$ |

**Additional DG challenges from FL.** For further understanding, we explore some additional questions on several smaller datasets because it is computationally feasible to train many different models on these datasets. Specifically, we explore how the number of clients, amount of communication (i.e., the number of server aggregations) in the federated setup, and client heterogeneity affects the performance of various methods. The figures and detailed analysis are provided in the Appendix but we highlight two remarks here.

*Remark* 4.7. **The number of clients $C$ strongly influences performance**. Performance in DG plunges from $90\%$ to as low as $10\%$ as $C$ shifts from 1 to 200 as suggested in Fig. 2. We explore this on 4 representative methods on three datasets. We advocate for future Federated DG assessments to consider larger number of clients, like 50 to 100 rather than only considering small numbers of clients. This demonstrates pressing, unaddressed challenges in Federated DG when grappling with many clients. Notably, FedADG and FedSR degrade faster with increasing number of clients.

*Remark* 4.8. **The number of communications does not monotonically affect the DG performance.** In the FL context, we note a unique implicit regularization phenomenon: optimal performance is achieved in relatively few communication rounds Fig. 3. For instance, with PACS, the ideal communication round is just 10 (while maintaining constant computation). Contrarily, for in-domain tasks, FL theory indicates that increased communication enhances performance, as shown by (Stich, 2018, Theorem 5.1). Investigating how DG accuracy relates to communication rounds, and potential implicit regularization through early stopping, offers a compelling direction for future research.

## 5 CONCLUSION AND DISCUSSION

We've established a benchmark methodology for evaluating DG tasks in the FL regime, assessing 14 prominent methods on 7 versatile datasets. Our findings indicate that Federated DG remains an unresolved challenge, with detailed gaps outlined in appendix Table 12. **Recommendations for future evaluations of Federated DG. 1)** Evaluation on the dataset where $R_{\text{DG}} < 1$. If $R_{\text{DG}} \approx 1$, there will be no real train-test heterogeneity. **2)** The FL regime introduces two main challenges for domain generalization (DG) methods. Firstly, the complete heterogeneity scenario limits the exchange of information between domains. Secondly, when ($R_{\text{FL}} < 1$), it poses convergence challenges for the methods. One way to produce lower $R_{\text{FL}}$ is to increase the number of clients. In future evaluations, it would be valuable to assess the capabilities and limitations of proposed methods in handling these challenges. **3)** Always include FedAvg as the baseline comparison. **4)** Evaluation on both Subpopulation shift datasets and general DG datasets since the methods might perform differently. **Suggestions for future work in Federated DG. 1)** The federated domain generalization (DG) task remains unsolved, and further work is needed to address the challenges. **2)** While our primary focus remains on domain generalization (DG), it is important for future methods to also address the issue of improving convergence when the FL learning rate ($R_{\text{FL}}$) is small. This consideration holds particular significance in real-world applications where efficient convergence is crucial. We hope this work provide a better foundation for future work in Federated DG and spur research progress.

## REPRODUCIBILITY STATEMENT

Code for reproduce the results is available at the following link:

https://github.com/inouye-lab/FedDG_Benchmark.

We include detailed documentation in using our code to reproduce the results throughout the paper. We also provide documentation in adding new algorithm's DG evaluation in the FL context. The code contains all the experiments included in section 4. To help reproducibility, we have

1. Include the requirements.txt file generated by conda to help environment setup.
2. The code contains a detailed Readme file to help code reading, experiment reproducing and future new method implementing.
3. We include the config file identical to the paper including the seed to help reproducibility.
4. We include the command needed to run the experiments.
5. The model structure and tokenizers are available.
6. We include the hyperparameter grid search configuration to reproduce Table 7.
7. The dataset preprocessing scripts, transform functions are also included.

We hope our reproducible and extendable codebase could help both new methods implementations and better evaluation in the Federated DG fields.

## ACKNOWLEDGEMENT

This work was supported by Army Research Lab under Contract No. W911NF-2020-221. R.B. and D.I. also acknowledge support from NSF (IIS-2212097) and ONR (N00014-23-C-1016). Any opinions, findings, and conclusions or recommendations expressed in this material are those of the authors and do not necessarily reflect the views of the sponsor(s).

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

# Appendix

## Table of Contents

## A   DATA PARTITION IN FEDERATED DG

### A.1   HETEROGENEOUS PARTITIONING ALGORITHM AND ITS GUARANTEE

Denote $\mathcal{P}_0$ as the complete heterogeneous case corresponding to Heterogeneous Partitioning with $\lambda = 0$. We have the following guarantees on the optimality of $\mathcal{P}_0$.

**Proposition A.1.** *When $C \geq D$, $\mathcal{P}_0$ is optimal for Eqn. 4. When $C < D$, Eqn. 4 is NP hard, and $\mathcal{P}_0$ is a fast greed approximation.*

*Proof.* **Case I.** When $C \geq D$, $\mathcal{P}_0$ generates $\{S_c\}, c \in [C]$ such that for all clients which share a same domain, their sample size is the same, that is, for all $c, c \in C'$ share some domain $d \in D$, there holds

$$n_c = n_{c'}. \tag{8}$$

Further, for sample variance, there holds

$$
\frac{1}{C-1} \sum_{c=1}^{C} \|n_c - \bar{n}\|_2^2
$$

$$
= \frac{1}{2C(C-1)} \sum_{i=1}^{C} \sum_{j=1}^{C} \|n_i - \bar{n} - n_j + \bar{n}\|_2^2
$$

$$
= \frac{1}{2C(C-1)} \sum_{i=1}^{C} \sum_{j=1}^{C} \|n_i - n_j\|_2^2. \tag{9}
$$

Therefore, Eqn. 4 boils down to

$$\widehat{\mathcal{P}} \in \underset{\mathcal{P} \in C_1 \cap C_2}{\arg\min} \frac{1}{2C(C-1)} \sum_{i=1}^{C} \sum_{j=1}^{C} \|n_i - n_j\|_2^2. \tag{10}$$

Chaining with Eqn. 8, we have

$$\frac{1}{2C(C-1)} \sum_{i=1}^{C} \sum_{j=1}^{C} \|n_i - n_j\|_2^2 = \frac{1}{2C(C-1)} \sum_{i=1}^{D} \sum_{j=1}^{D} \|\tilde{n}_i - \tilde{n}_j\|_2^2, \tag{11}$$

where $\tilde{n}_i$ is the same sample size of clients which share the domain $i$. The equality comes from the second equation in Eqn. 6, where we reassign the same sample sizes for those $c' \in \{1, \ldots, c\}$ which share the domain $d^*$ as

$$\frac{n_{d^*}}{\sum_{c'=1}^{C} \mathbb{1}[d^* \in \mathcal{D}_{c'}]}. \tag{12}$$

In addition, notice that we always divide the currently largest average domain $d^*$ between the new client and the original clients associated with it. That is,

$$d^* \in \underset{d \in [D]}{\arg\max} \frac{n_d}{\sum_{c'=1}^{C} \mathbb{1}[d \in \mathcal{D}_{c'}]}. \tag{13}$$

Therefore, the discrepancy between $\tilde{n}_i$ and $\tilde{n}_j$, $i, j \in [D]$ is minimized. It is straightforward to verify such partition $\mathcal{P}_0$ also satisfy the true partition (constraint). Thus, $\mathcal{P}_0$ is optimal for Eqn. 4.

**Case II.** When $C < D$, the problem corresponds to the classic NP-Hard problem known as Multiway Number Partitioning. We employ a well-known linear-time greedy algorithm to address this (Graham, 1969).

□

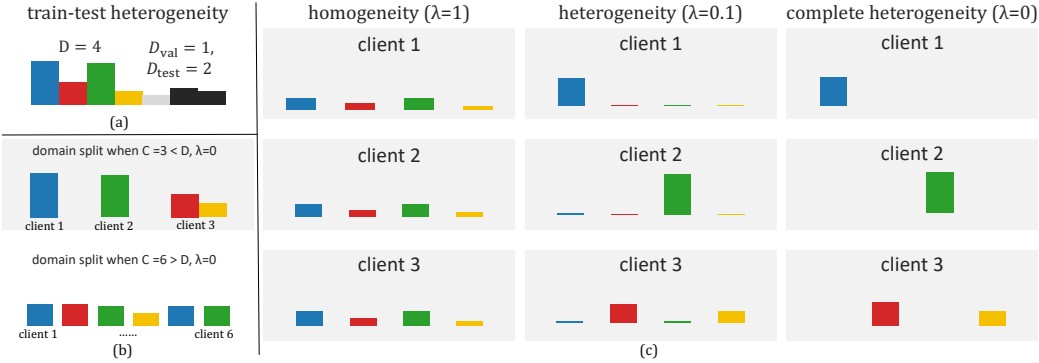

Figure 1: Distinct color refers to distinct domain data, and $\lambda$ is the domain balancing parameter. (a): train-test domain heterogeneity. (b): domain partitioning when $C \leq D$ and domain partitioning when $C > D$. (c): domain partitioning illustration when $C \leq D$; homogeneous ($\lambda = 1$), heterogeneous ($\lambda = 0.1$), and extreme heterogeneous ($\lambda = 0$).

We provide the pseudo code for Heterogeneous Partitioning in Alg. 1 and an illustration in Fig. 1. In Fig. 1, (a) represents a multi-domain dataset with 4 training domains. Each color represents a domain and the height of the bar represents the number of samples in this domain. (b) shows the complete heterogeneity with two scenarios $C < D$ and $C > D$. (c) is a detailed partition result with different interpolating with the $C < D$ case. It can be seen that in the complete heterogeneity case, the partition satisfies constraint $1, 2$, further, the sample size between clients are roughly even. For other cases, True Partition (Constraint) is satisfied and also the sample size between clients are roughly even.

## A.2  OTHER PARTITION METHODS

**Shards.** By adjusting the constant from $2$ to other value, it's possible to regulate the heterogeneity to a certain extent. However, if the dataset contains imbalance number of samples per domain, shards method could not achieve the complete heterogeneity. For instance, if a dataset contains 2 domains with $20, 30$ data samples respectively. Shard partitioning will not be able to assign each client with data from only one domain.

**Dirichlet Partitioning.** It draws $q_c \sim \text{Dir}(\alpha p)$, where $p$ characterizes a prior domain distribution over $D$ domains for each client $c \in [C]$. $\alpha > 0$ is a concentration parameter controlling the heterogeneity among clients. As $\alpha \to 0$, each client predominantly holds samples of a single label. Conversely, as $\alpha \to \infty$, the distribution of labels for each client becomes identical. This method is not a desirable partitioning in the following sense. **1)** If sampling with replacement, then the datasets across clients may not be disjoint, and some data samples may not be allocated to any client, therefore breaking properties 3 and 4. **2)** If sampling without replacement, the method is inapplicable when the available data from one domain doesn't meet the collective demand for this domain's data from all clients, breaking property 2.

**Semantic Partitioning.** This method aims to partition dataset based on the feature similarity without domain labels. It first processes the data using a powerful pretrained model. The outputs before the fully connected layer are later used as features. Then, thses features are fit a Gaussian Mixture Model to form $K$ for each clusters for each class $Y$. These $KY$ clusters are then merged iteratively using an optimal bipartite for random label pairs across different class. This

---

**Algorithm 1** Domain Partition Algorithm

**Input** Number of clients $C$, heterogeneity parameter $\lambda \in [0, 1]$, and samples from each domain $(\mathcal{S}_1, \cdots, \mathcal{S}_D)$, where $n_1, \ldots, n_D$ denote the number of samples per domain and w.l.o.g. are assumed to be in descending order, i.e., $n_1 \geq n_2 \geq \cdots \geq n_D$.

$\quad$ *// Divide domain indices across clients*
$\quad$ **if** $C \leq D$ **then**
$\quad\quad$ $\forall c, \mathcal{D}_c \leftarrow \emptyset$
$\quad\quad$ **for** $d = 1, 2, \ldots, D$ **do**
$\quad\quad\quad$ *// Find client with fewest samples*
$\quad\quad\quad$ $c^* \in \underset{c \in [C]}{\arg\min} \sum_{d' \in \mathcal{D}_c} n_{d'}$
$\quad\quad\quad$ $\mathcal{D}_{c^*} \leftarrow \mathcal{D}_{c^*} \cup \{d\}$
$\quad\quad$ **end for**
$\quad$ **else if** $C > D$ **then**
$\quad\quad$ $\forall c \in \{1, 2, \ldots, D\}, \mathcal{D}_c \leftarrow \{c\}$
$\quad\quad$ **for** $c = D + 1, \ldots, C$ **do**
$\quad\quad\quad$ *// Find on average largest domain to partition*
$\quad\quad\quad$ $d^* \in \underset{d \in [D]}{\arg\max} \frac{n_d}{\sum_{c'=1}^{C} \mathbb{1}[d \in \mathcal{D}_{c'}]}$
$\quad\quad\quad$ $\mathcal{D}_c \leftarrow \{d^*\}$
$\quad\quad$ **end for**
$\quad$ **end if**
$\quad$ *// Partition samples across the clients*
$\quad$ $\forall c, \mathcal{S}_c \leftarrow \emptyset$
$\quad$ **for** $c \in [C], d \in [D]$ **do**
$\quad\quad$ $n_{d,c}(\lambda) = \lambda \frac{n_d}{C} + (1 - \lambda) \frac{\mathbb{1}[d \in \mathcal{D}_c] \cdot n_d}{\sum_{c'=1}^{C} \mathbb{1}[d \in \mathcal{D}_c]}$
$\quad\quad$ $\mathcal{S}' \leftarrow \texttt{SampleWOReplacement}(\mathcal{S}_d, n_{d,c})$

$\quad\quad$ $\mathcal{S}_d \leftarrow \mathcal{S}_d \setminus \mathcal{S}'$ $\quad$ *// Remove from domain*
$\quad\quad$ $\mathcal{S}_c \leftarrow \mathcal{S}_c \cup \mathcal{S}'$ $\quad$ *// Append to client dataset*
$\quad$ **end for**
**Output** $(\mathcal{S}_1, \mathcal{S}_2, \cdots, \mathcal{S}_C)$

---

partition method is more heuristic and highly rely on the quality of the pretrained model. Furthermore, since there is no hard code label, and the partition is based on KL divergence, this method could not give a complete heterogeneity which violate property 1. Despite that this method could not guarantee property 1, it is a good partition method to create "domain"-heterogeneity without having domain labels.

*Example* 1. We provide a toy example of partition a dataset containing 5 training domains with $10, 20, 30, 40, 100$ data samples respectively. We illustrate how shards and Dirichlet methods work and highlight that disjointedness flexibility and controllability are violated when our goal is to partition the data into 2 clients. Shards partitioning might allocation $[10, 20, 20, 0, 50]$ samples to the first client, and $[0, 0, 10, 40, 50]$ samples to the second client. This lead to the share of domain 3 and 5 between the two clients showing the violation of Complete Heterogeneity (Constraint). The Dirichlet partitioning will sample a distribution from a Dirichlet distribution with $\alpha \approx 0$. However, no matter what distribution are sampled, at least 3 domains will be neglected, leading to the violation of True Partition (Constraint). In contrast, our Heterogeneous Partitioning will allocate $[10, 20, 30, 40, 0]$ to the first client and $[0, 0, 0, 0, 100]$ to the second client, effectively maintaining Complete Heterogeneity (Constraint), True Partition (Constraint), in addition, the sample size difference is 0 in this case.

## B CURRENT METHODS IN SOLVING DG

**Centralized DG methods**  Most work solving in DG lies in the centralize regime. A predominant and effective centralized DG approach is through representation learning. Arjovsky et al. (2019) tries to learn domain-invariant feature by aligning the conditional distribution of $p(Y|X)$ among different domains. Sun and Saenko (2016), and Li et al. (2018) tries to explicitly align the first order and second order momentum in the feature space. There are also methods trying to promote the out-of-distribution generalization by posting constraint on the gradient information among different domains, where Shi et al. (2021) tries to align the gradient among different domains, and Rame et al. (2022) enforces domain invariance in the space of the gradients of the loss. Other approaches include distributionally robust optimization (Sagawa et al., 2019), where this method learns the worst-case distribution scenario of training domains. Furthermore, Zhang et al. (2017) is not specifically design for solving domain generalization, but found to be effective as well when we extrapolate the data from different domains in the training dataset. Xu et al. (2020) introduces domain-mixup which extrapolates examples and features across training domains.

**Federated DG methods**  Limited research has focused explicitly on solving the Federated DG by design. FedDG Liu et al. (2021) introduced a specific FL paradigm for medical image classification, which involves sharing the amplitude spectrum of images among local clients, violating the privacy protocol. Another approach, FedADG (Zhang et al., 2021), utilizes a generative adversarial network (GAN) framework in FL, where each client contains four models: a featurizer, a classifier, a generator, and a discriminator. FedADG first trains the featurizer and classifier using empirical loss and then trains the generator and discriminator using the GAN approach. However, this method requires training four models simultaneously and tuning numerous hyperparameters, making convergence challenging. A novel aggregation method called Federated Gradient Masking Averaging (FedGMA) (Tenison et al., 2022b) aims to enhance generalization across clients and the global model. FedGMA prioritizes gradient components aligned with the dominant direction across clients while assigning less importance to inconsistent components. FedSR (Nguyen et al., 2022) proposes a simple algorithm that uses two locally-computable regularizers for domain generalization. Given the limited literature on solving domain generalization (DG) in the federated learning (FL) setting, we selected all the aforementioned algorithms.

**FL methods tackling client heterogeneity**  Another line of research in FL aims to guarantee convergence even under client heterogeneity, but these FL-based methods still assume the train and test datasets do not shift (i.e., they do not explicitly tackle train-test heterogeneity of the domain generalization task). We include this line of work because methods considering converging over a heterogeneous dataset might bring better DG ability implicitly. The empirical observation of the statistical challenge in federated learning when local data is non-IID was first made by Zhao et al. (2018). Several subsequent works have analyzed client heterogeneity by assuming bounded gradients (Yu et al., 2019; Basu et al., 2019; Wang et al., 2019; Li et al., 2019) or bounded gradient dissimilarity (Li et al., 2020), and additionally assuming bounded Hessian dissimilarity (Karimireddy et al., 2020; Khaled et al., 2020; Liang et al., 2019). From this category, we selected FedProx (Li et al., 2020), which addresses statistical heterogeneity by adding a proximal term to the local subproblem, constraining local updates to be closer to the initial global model. Scaffold (Karimireddy et al., 2020) utilizes variance reduction to account for client heterogeneity.

---

**Algorithm 2** Modified AFL

**Initialize** $\theta$, $\beta$

    Transmit $\theta$ to all clients $c \in [C]$.
    **for** $t = 1, 2, \ldots, T$ **do**
        // ———Update $\theta$———
        Transmit $\beta$ to all clients $c$.
        **for** $c = 1, 2, \ldots, C$ **do**
            Update $\theta_c$ using SGD with multiple iteration.
            Submit $\theta_c$ to the server.
        **end for**
        $\theta \leftarrow \sum_c \frac{1}{n_c} \theta_c$.     // Weighted average.
        // ———Update $\beta$———
        Transmit $\theta$ to all clients $c$
        **for** $c = 1, 2, \ldots, C$ **do**
            Calculate $\ell_{d,c} = \nabla_{\beta_d} \mathcal{L}_c$
            Submit updated $\ell_{d,c}$.
        **end for**
        **for** $d = 1, 2, \ldots, D$ **do**
            $\beta_d \leftarrow \text{Proj}(\beta_d + \gamma_\beta \sum_c \ell_{d,c})$.
        **end for**
    **end for**
**Output** $\theta$

---

**FL methods tackling fairness** Agnostic Federated Learning (AFL) (Mohri et al., 2019; Du et al., 2021) shares similarities with Domain Generalization in a Federated context. This is evident as both approaches address scenarios where the test distribution diverges from the training distribution. AFL constructs a framework that naturally yields a notion of fairness, where the centralized model is optimized for any target distribution formed by a mixture of the client distributions. Thus, AFL is a good method to evaluate especially when tackling subpopulation shift tasks. AFL introduces a minimax objective which is identical to the GroupDRO (Sagawa et al., 2019),

$$\min_{\theta} \max_{\beta \in \Delta_D} \mathcal{L}(\theta, \beta) = \sum_{d=1}^{D} \beta_d \ell_d(\theta), \tag{14}$$

where $\theta$ denotes the model parameter and $\beta$ is a weight vector taking values in the simplex $\Delta_D$ of dimension $D$. AFL introduces an algorithm applying projected gradient ascent on $\beta$ and gradient descent on $\theta$. It is designed for $C = D$, where they assumes each client as a domain. Further, it requires communication per iteration. We made the following two modifications to accommodate the general $C \neq D$ cases and expensive communication cost:

1. To allow $C \neq D$, we construct new objective as the following:

$$\min_{\theta} \max_{\beta \in \Delta_D} \mathcal{L}(\theta, \beta) = \sum_{c=1}^{C} \mathcal{L}_c(\theta, \beta) = \sum_{c=1}^{C} \sum_{d=1}^{D} \beta_d \ell_{d,c}(\theta). \tag{15}$$

2. To reduce communication cost, we allow $\theta$ to be updated multiple iterations locally per communication. Further, we maintain the update of $\beta$ on the central server, as in AFL, given that it requires projection of the global updates onto the simplex. This projection is not equivalent to averaging the locally projected updates of $\beta$.

Modified AFL avoids the frequent communication compared to AFL.

## C   DATASETS AND DIFFICULTY METRIC

### C.1   DATASET INTRODUCTION

In this section, we introduce the datasets we used in our experiments, and the partition method we used to build heterogeneous datasets in the training and testing phase as well as the heterogeneous local training datasets among clients in the FL.

**FEMNIST** It is an FL prototyping image dataset of handwritten digits and characters each users created as a natural domains, widely used for evaluation for client heterogeneity in FL. Though it contain many training domains, it lacks significant distribution shifts across domains ($R_{\text{DG}} = 1$), and considered as easy compared to other datasets.

**PACS** It is an image dataset for domain generalization. It consists of four domains, namely Photo (1,670 images), Art Painting (2,048 images), Cartoon (2,344 images), and Sketch (3,929 images). This task requires learning the classification task on a set of objects by learning on totally different renditions. $R_{\text{DG}} = 0.960$ makes it a easy dataset as domain generalization in our setting. Notice that choosing different domain as test domain might give us different $R_{\text{DG}}$.

**IWildCam** It is a real-world image classification dataset based on wild animal camera traps around the world, where each camera represents a domain. It contains 243 training domains, 32 validation and 48 test domains. Usually people cares about rare speicies, thus we utilize the macro F1 score as the evaluation metric instead of standard accuracy, as recommended in the original dataset's reference (Koh et al., 2021). The $R_{\text{DG}} = 0.449$ makes it a very challenging dataset for domain generalization.

**CelebA** CelebA (Celebrity Attribute) (Liu et al., 2015) is a large-scale face attributes dataset. It's one of the most popular datasets used in the facial recognition and facial attribute recognition domains. We use a variant of the original CelebA dataset from (Sagawa et al., 2019) using hair color as the classification target and gender as domain labels. This forms a subpopulation shift task which is a special kind of domain generalization.

**Camelyon17** It is a medical dataset consisting of a set of whole-slide images from multiple institutions (as domain label) to detect breast cancer. It consist 3 training domains, 1 validation domain and 1 test domain. We partition the data following wilds benchmark (Koh et al., 2021).

**CivilComments** It is a real-world binary classification text-based dataset formed from the comments from different demographic groups of people, containing 8 demographic group. The goal is to judge whether the text is malicious or not, which is also a subpopulation shift dataset.

**Py150** It is a real-world code-completion dataset which is challenging given massive domains 8421, where the domain is formed according to the repository author. The goal is to predict the next token given the context of previous tokens. We evaluate models by the accuracy on the class and method tokens.

Table 6: Summary of selected datasets as well as their difficulty metric, where $n$ is the number of samples, $D$ is the number of domains, $C$ is the number of clients used.

| Dataset | Statistics | | | Difficulty Metric | | | |
| | $n$ | $D$ | $C$ | $R_{DG}$ | $R_{FL}(\lambda)$ | | |
| | | | | | $\lambda = 1$ | $\lambda = 0.1$ | $\lambda = 0$ |
| --- | --- | --- | --- | --- | --- | --- | --- |
| FEMNIST | 737036 | 2586 | 100 | 1.000 | 0.980 | 0.981 | 0.981 |
| PACS | 9991 | 2 | 100 | 0.960 | 1.000 | 1.000 | 1.000 |
| IWildCam | 203029 | 243 | 100 | 0.449 | 0.869 | 0.714 | 0.571 |
| CelebA | 162770 | 4 | 100 | 0.661 | 0.760 | 0.805 | 0.797 |
| Camelyon17 | 410359 | 3 | 100 | 0.969 | 1.000 | 1.000 | 1.000 |
| CivilComments | 448000 | 16 | 100 | 0.984 | 0.618 | 0.533 | 0.532 |
| Py150 | 150000 | 8421 | 100 | 0.969 | 0.999 | 0.998 | 0.998 |

### C.2 DATASET PARTITION SETUP

For each dataset, we first partition the dataset into 5 categories, namely training dataset, in-domain validation dataset, in-domain test dataset, held-out validation dataset and test domain dataset. For FEMNIST and PACS dataset, we use Cartoon and Sketch as training domain, Art-painting as held-out domain, Painting as test domain. For training domain, we partition $10\%, 10\%$ of the total training domain datasets as in-domain validation dataset and in-domain test dataset respectively. For IWildCam, CivilComments and Py150, we directly apply the Wilds official partitions.

## D BENCHMARK EXPERIMENTAL SETTING

### D.1 MODEL STRUCTURE

In this benchmark, for image-based datasets, we use ResNet-50 He et al. (2016) dataset. For CivilComments and Py150 dataset, we choose DistilBERT Sanh et al. (2020) and CodeGPT Radford et al. (2019) respectively as recommended by Wilds (Koh et al., 2021).

### D.2 MODEL SELECTION

We conduct held-out domain model selection with 4 runs for each methods. The oracle model selection evaluates the model based on the performance on the held-out validation domain. The results are reported based on the best run.

### D.3 EARLY STOPPING

We conduct early stopping using the held-out validation dataset in our evaluation. For each dataset and method, We first run certain communication rounds, then we select the model parameters which achieves the best performance on the validation set. We report the held-out validation dataset in the main paper, and we report the results using the in-domain validation set in Appendix F.

### D.4 HYPERPARAMETERS

In this section, we present the hyperparameters selected for the evaluation. We grid search 8 times of run starting with learning rate same as ERM and other hyperparameters from the methods' original parameters. We than select the hyperparameter based on the best performance on the held-out domain validation set. Please refer to the Table 7 to review all hyperparameters.

Table 7: Hyperparameters used in the evaluation.

| Dataset Name Total Communication | | PACS 80 | CelebA 20 | Camelyon17 20 | FEMNIST 40 | IWildCam 50 | CivilComments 10 | Py150 10 |
|---|---|---|---|---|---|---|---|---|
| ERM | lr | $3 \times 10^{-5}$ | $3 \times 10^{-3}$ | $1 \times 10^{-3}$ | $1 \times 10^{-3}$ | $3 \times 10^{-5}$ | $1 \times 10^{-5}$ | $8 \times 10^{-5}$ |
| IRM | lr | $3 \times 10^{-5}$ | $1 \times 10^{-3}$ | $1 \times 10^{-3}$ | $1 \times 10^{-3}$ | $3 \times 10^{-5}$ | $1 \times 10^{-6}$ | $1 \times 10^{-6}$ |
| | penalty weight | 1000 | 1000 | 1000 | 100 | 100 | 10 | 1000 |
| Fish | lr | $3 \times 10^{-5}$ | $1 \times 10^{-3}$ | $1 \times 10^{-4}$ | $1 \times 10^{-3}$ | $1 \times 10^{-5}$ | $1 \times 10^{-6}$ | $1 \times 10^{-6}$ |
| | meta-lr | 0.1 | 0.1 | 0.1 | 0.1 | 0.1 | 0.1 | 1 |
| Mixup | lr | $3 \times 10^{-5}$ | $3 \times 10^{-4}$ | $1 \times 10^{-3}$ | $1 \times 10^{-3}$ | $3 \times 10^{-5}$ | - | - |
| | $\alpha$ | 0.2 | 0.2 | 0.1 | 0.2 | 0.2 | - | - |
| MMD | lr | $3 \times 10^{-5}$ | 0.0001 | $1 \times 10^{-3}$ | $1 \times 10^{-5}$ | $3 \times 10^{-5}$ | $1 \times 10^{-6}$ | $1 \times 10^{-5}$ |
| | penalty weight | 1 | 1 | 0.1 | 1 | 1 | 10 | 1 |
| Coral | lr | $3 \times 10^{-5}$ | $1 \times 10^{-4}$ | $1 \times 10^{-3}$ | $1 \times 10^{-5}$ | $3 \times 10^{-5}$ | $1 \times 10^{-6}$ | $5 \times 10^{-5}$ |
| | penalty weight | 1 | 1 | 10 | 10 | 1 | 1 | 1 |
| GroupDRO | lr | $3 \times 10^{-5}$ | $3 \times 10^{-4}$ | $1 \times 10^{-3}$ | $1 \times 10^{-5}$ | $1 \times 10^{-5}$ | $1 \times 10^{-5}$ | $1 \times 10^{-5}$ |
| | group lr | 0.01 | 0.05 | 0.01 | $1 \times 10^{-5}$ | $1 \times 10^{-5}$ | 0.05 | 0.005 |
| FedProx | lr | $3 \times 10^{-5}$ | $1 \times 10^{-3}$ | $5 \times 10^{-4}$ | $1 \times 10^{-5}$ | $3 \times 10^{-5}$ | $1 \times 10^{-5}$ | $8 \times 10^{-6}$ |
| | $\mu$ | $1 \times 10^{-3}$ | 0.1 | $1 \times 10^{-3}$ | $1 \times 10^{-3}$ | 0.1 | 0.01 | $1 \times 10^{-3}$ |
| Scaffold | lr | $3 \times 10^{-5}$ | $1 \times 10^{-4}$ | $1 \times 10^{-3}$ | $1 \times 10^{-5}$ | $1 \times 10^{-5}$ | $3 \times 10^{-6}$ | $8 \times 10^{-6}$ |
| AFL | lr | $3 \times 10^{-5}$ | $1 \times 10^{-4}$ | $1 \times 10^{-3}$ | $1 \times 10^{-5}$ | $1 \times 10^{-5}$ | $3 \times 10^{-6}$ | $8 \times 10^{-6}$ |
| FedDG | lr | $3 \times 10^{-5}$ | $1 \times 10^{-4}$ | $1 \times 10^{-4}$ | $1 \times 10^{-4}$ | $1 \times 10^{-4}$ | - | - |
| | $\lambda$ | $U(0,1)$ | $U(0,1)$ | $U(0,1)$ | $U(0,1)$ | $U(0,1)$ | - | - |
| FedADG | Classifier lr | $2 \times 10^{-4}$ | $2 \times 10^{-4}$ | $2 \times 10^{-4}$ | $2 \times 10^{-4}$ | $2 \times 10^{-4}$ | - | - |
| | Gen lr | $7 \times 10^{-4}$ | $5 \times 10^{-4}$ | $7 \times 10^{-4}$ | $1 \times 10^{-3}$ | $1 \times 10^{-5}$ | - | - |
| | Disc lr | $7 \times 10^{-4}$ | $5 \times 10^{-4}$ | $7 \times 10^{-4}$ | $1 \times 10^{-3}$ | $1 \times 10^{-5}$ | - | - |
| | $\alpha$ | 0.15 | 0.25 | 0.1 | 0.05 | 0.2 | - | - |
| FedSR | lr | $1 \times 10^{-5}$ | $5 \times 10^{-4}$ | $1 \times 10^{-4}$ | $5 \times 10^{-4}$ | $3 \times 10^{-5}$ | $5 \times 10^{-6}$ | $5 \times 10^{-6}$ |
| | l2 regularizer | 0.01 | 0.01 | 0.01 | 0.01 | 0.01 | 0.01 | 0.01 |
| | cmi regularizer | 0.001 | 0.001 | 0.001 | 0.001 | 0.001 | 0.001 | 0.001 |
| FedGMA | lr | $3 \times 10^{-5}$ | $1 \times 10^{-4}$ | $5 \times 10^{-4}$ | $1 \times 10^{-3}$ | $1 \times 10^{-3}$ | $5 \times 10^{-6}$ | $1 \times 10^{-5}$ |
| | mask-threshold | 0.1 | 0.1 | 0.4 | 0.4 | 0.4 | 0.4 | 0.4 |

## E ADDITIONAL FL-SPECIFIC CHALLENGES FOR DOMAIN GENERALIZATION

As mentioned in subsection 4.3, we also include some deeper exploration over the effect of number of clients and communication frequency, which are unique to the FL regime.

**i) Massive number of clients:** In this experiment, we explore the performance of different algorithms when the number of clients $C$ increases on PACS. We fix the communication rounds 50 and the local number of epoch is 1 (synchronizing the models every epoch). Fig. 2 plots the held-out DG test accuracy versus number of clients for different levels of data heterogeneity. The following comments are in order: given communication budget, 1) current domain generalization methods all degrade a lot after $C \geq 10$, while the performance ERM and FedDG maintain relatively unchanged as the clients number increases given communication budget. FedADG and FedSR are are sensitive to the clients number, and they both fail after $C \geq 20$. 2) Even in the simplest homogeneous setting $\lambda = 1$, where each local client has i.i.d training data, current domain generalization methods IRM, FISH, Mixup, MMD, Coral, GroupDRO work poorly in the existence of large clients number, this means new methods are needed for DG in FL context when data are stored among massive number of clients.

**ii) Communication constraint:** To show the effect of communication rounds on convergence, we plot the test accuracy versus communication rounds in Appendix Fig. 3. We fix the number of clients $C = 100$ on PACS and decreases rounds of communication (together with increasing local epochs).

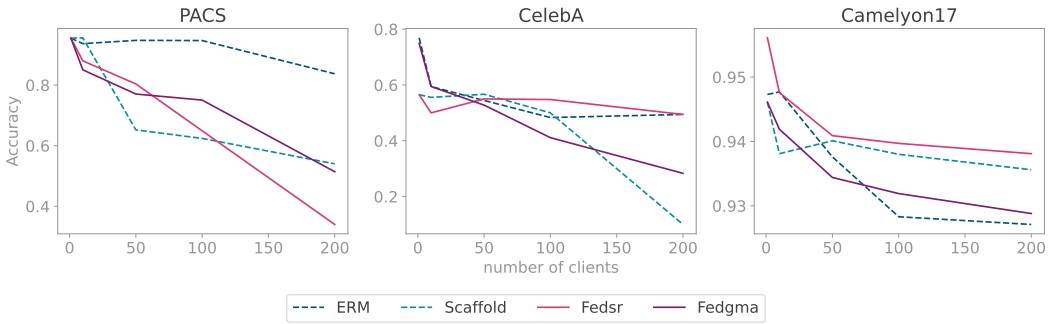

Figure 2: Performance based on accuracy versus the number of clients across the PACS, CelebA, and Camelyon17 datasets.

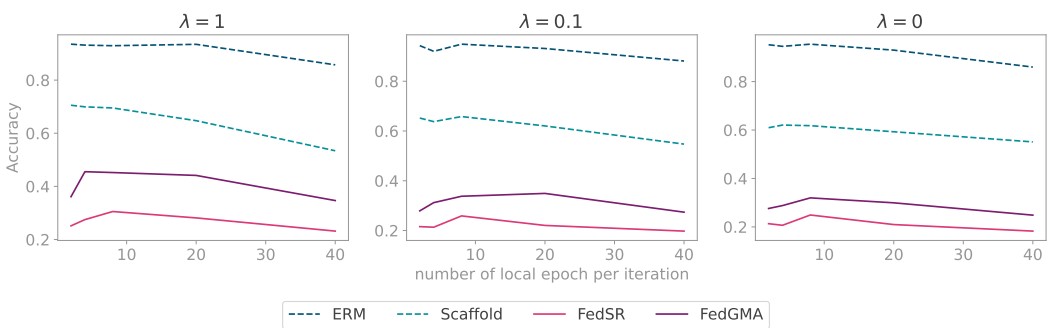

Figure 3: PACS: Held-out DG test accuracy vs. varying communications (resp. varying echoes ).

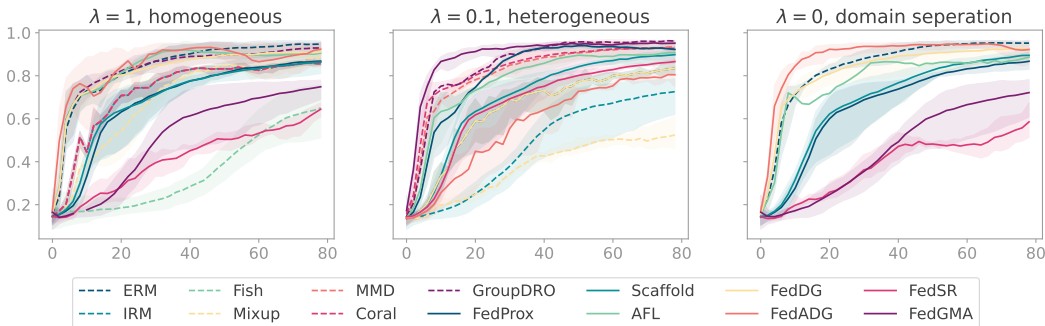

Figure 4: Convergence curve on PACS; total clients and training domains $(C, D) = (100, 2)$; increasing domain heterogeneity from left to right: $\lambda = (1, 0.1, 0)$.

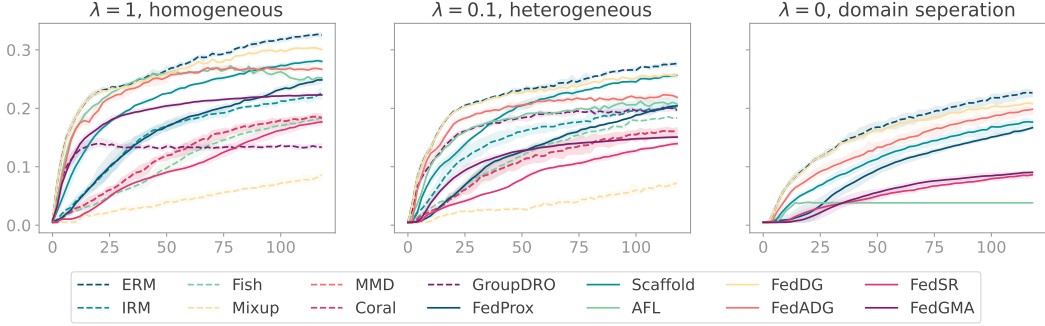

Figure 5: Accuracy versus communication rounds for IWildCam; Total clients number $C = 243$; increasing heterogeneity from left to right panel: $\lambda = (1, 0.1, 0)$.

That is, if the regime restricts the communication budget, then we increase its local computation $E$ to have the same the total computations. Therefore, the influence of communication on the performance is fair between algorithms because the total data pass is fixed. We observe that the total number of communications does not monotonically affect the DG performance. With decreasing number of total communication, most methods' performance first increase then decrease. This might be an interesting implicit regularization phenomena in the FL context. Without discussing DG task, usually frequent communications lead to faster convergence. The relationship between DG performance and communications requires further exploration.

## F    SUPPLEMENTARY RESULTS

In the main paper, we provide experiments results using held-out validation early stopping. Here we report the results using in-domain validation set for reference. We also report the convergence curve for each methods on each dataset for reference. We observe that under most of cases, the held-out validation gives us a better model. Thus, we recommend using held-out validation set to perform early stopping when we can access multiple training domains. However, if the training dataset contains only 2-3 domains, we should consider using in-domain validation set.

**More results on PACS and IWildCam dataset.**

**Results on FEMNIST dataset.**    As mentioned in the main paper, we include the FEMNIST dataset here for reference. It is evident from Table 11 that $\lambda$ does not significantly influence the final domain generalization (DG) accuracy, and whether using in-domain validation or held-out-domain validation does not impact the final DG accuracy as well. This suggests a lack of statistical heterogeneity across different domains. Furthermore, we observe that changing $\lambda$ does not significantly affect the convergence. Most of the results do not converge to the performance level of the centralized counterpart. This discrepancy arises from the challenge posed by the large number of clients, where $R_{\text{FL}} = 0.980 < 1$.

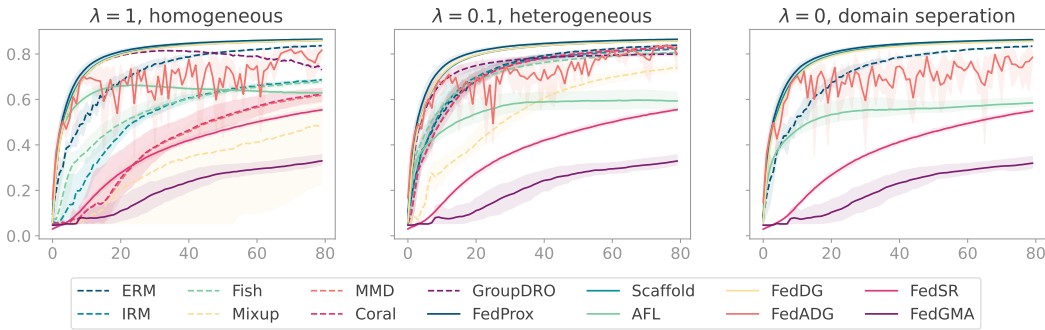

Figure 6: Accuracy versus communication rounds for FEMNIST; total clients and training domains $(K, M) = (100, 2586)$; increasing domain heterogeneity from left to right panel: $\lambda = (1, 0.1, 0)$.

## G    GAP TABLE

We list the gap table in Table 12 for summarizing the current DG algorithms performance gap w.r.t FedAvg-ERM in the FL context, in particular, positive means it outperforms FedAvg-ERM, negative means it is worse than FedAvg-ERM. It can be seen that in the on the simple dataset, the best DG migrated from centralized setting is better than FedAvg-ERM. In the complete heterogeneity case, no centralized DG algorithms can be adapted to it, and FDG methods performs comparably good in this setting. However, they fail in harder datasets. In the hardest setting, currently the Federated methods dealing with data heterogeneity performs the best. It is worth noting that while federated learning methods that address client heterogeneity perform better than other methods, they still fall short of achieving centralized empirical risk minimization (ERM). This highlights the need for future research and development of DG methods in the FL regime.

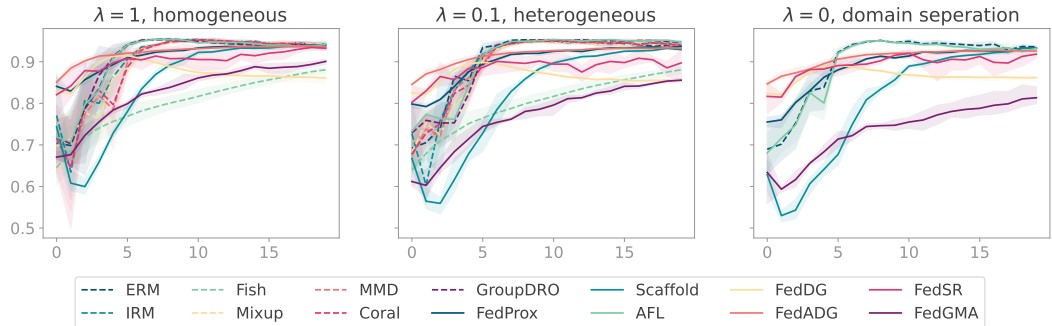

Figure 7: Accuracy versus communication rounds for Camelyon17; total clients and training domains $(K, M) = (100, 4)$; increasing domain heterogeneity from left to right panel: $\lambda = (1, 0.1, 0)$.

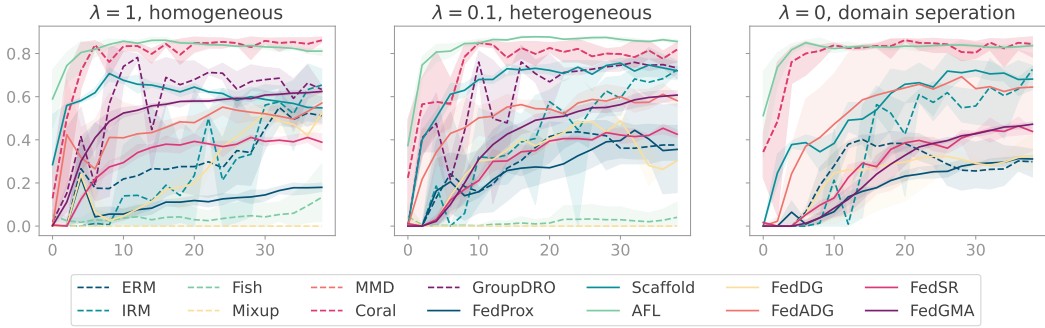

Figure 8: Accuracy versus communication rounds for CelebA; total clients and training domains $(K, M) = (100, 2)$; increasing domain heterogeneity from left to right panel: $\lambda = (1, 0.1, 0)$.

Table 8: Test accuracy on PACS and IWildCam dataset with held-out-domain validation where FedAvg-ERM is the simple baseline (B). "-" means the method is not applicable in that context. Bold is for best and italics is for second best in each column. We report the standard deviation among 3 runs.

| | | PACS ($D=2$) | | | | IWildCam ($D=243$) | | | |
|---|---|---|---|---|---|---|---|---|---|
| | | $C=1$ | $C=100$ (FL) | | | $C=1$ | $C=100$ (FL) | | |
| | | (centralized) | $\lambda=1$ | $\lambda=0.1$ | $\lambda=0$ | (centralized) | $\lambda=1$ | $\lambda=0.1$ | $\lambda=0$ |
| B | FedAvg-ERM | **0.943 ± 0.012** | **0.948 ± 0.020** | **0.955 ± 0.016** | **0.954 ± 0.018** | *0.343 ± 0.003* | **0.298 ± 0.002** | 0.245 ± 0.002 | **0.196 ± 0.010** |
| DG Adapted | IRM | 0.918 ± 0.007 | 0.920 ± 0.041 | 0.856 ± 0.044 | - | 0.321 ± 0.004 | 0.196 ± 0.006 | 0.179 ± 0.005 | - |
| | Fish | *0.936 ± 0.021* | 0.653 ± 0.127 | 0.345 ± 0.107 | - | **0.354 ± 0.001** | 0.183 ± 0.004 | 0.154 ± 0.005 | - |
| | Mixup | 0.918 ± 0.006 | 0.886 ± 0.052 | 0.826 ± 0.030 | - | 0.331 ± 0.007 | 0.086 ± 0.010 | 0.072 ± 0.007 | - |
| | MMD | 0.928 ± 0.022 | 0.921 ± 0.041 | 0.855 ± 0.044 | - | 0.324 ± 0.002 | 0.185 ± 0.005 | 0.163 ± 0.007 | - |
| | DeepCoral | 0.928 ± 0.007 | 0.920 ± 0.041 | 0.856 ± 0.044 | - | 0.329 ± 0.008 | 0.185 ± 0.005 | 0.163 ± 0.007 | - |
| | GroupDRO | 0.928 ± 0.007 | *0.944 ± 0.025* | 0.945 ± 0.015 | - | 0.211 ± 0.002 | 0.134 ± 0.005 | 0.198 ± 0.006 | - |
| FL | FedProx | - | 0.896 ± 0.025 | 0.891 ± 0.025 | 0.896 ± 0.024 | - | 0.251 ± 0.003 | 0.204 ± 0.001 | 0.167 ± 0.003 |
| | Scaffold | - | 0.896 ± 0.025 | 0.892 ± 0.024 | 0.896 ± 0.024 | - | *0.281 ± 0.002* | **0.255 ± 0.004** | *0.178 ± 0.008* |
| | AFL | - | 0.931 ± 0.026 | 0.915 ± 0.040 | 0.911 ± 0.041 | - | 0.256 ± 0.010 | 0.162 ± 0.008 | 0.034 ± 0.001 |
| FDG | FedDG | - | 0.933 ± 0.020 | *0.947 ± 0.018* | *0.943 ± 0.023* | - | 0.274 ± 0.001 | 0.235 ± 0.002 | 0.167 ± 0.004 |
| | FedADG | - | 0.943 ± 0.011 | 0.942 ± 0.001 | 0.935 ± 0.011 | - | 0.259 ± 0.012 | *0.251 ± 0.010* | 0.164 ± 0.001 |
| | FedSR | - | 0.640 ± 0.158 | 0.548 ± 0.086 | 0.537 ± 0.092 | - | 0.177 ± 0.005 | 0.141 ± 0.005 | 0.086 ± 0.003 |
| | FedGMA | - | 0.750 ± 0.050 | 0.730 ± 0.087 | 0.724 ± 0.099 | - | 0.223 ± 0.001 | 0.151 ± 0.002 | 0.091 ± 0.003 |

Table 9: Test accuracy on CelebA and Camelyon17 dataset with held-out-domain validation where FedAvg-ERM is the simple baseline (B). "-" means the method is not applicable in that context. Bold is for best and italics is for second best in each column. We report the standard deviation among 3 runs.

| | | CelebA ($D=2$) | | | | Camelyon17 ($D=3$) | | | |
|---|---|---|---|---|---|---|---|---|---|
| | | $C=1$ | $C=100$ (FL) | | | $C=1$ | $C=100$ (FL) | | |
| | | (centralized) | $\lambda=1$ | $\lambda=0.1$ | $\lambda=0$ | (centralized) | $\lambda=1$ | $\lambda=0.1$ | $\lambda=0$ |
| B | FedAvg-ERM | 0.769 ± 0.035 | 0.606 ± 0.019 | 0.517 ± 0.039 | 0.446 ± 0.018 | 0.903 ± 0.009 | *0.949 ± 0.007* | *0.950 ± 0.004* | **0.948 ± 0.004** |
| DG Adapted | IRM | **0.891 ± 0.012** | 0.706 ± 0.048 | 0.737 ± 0.013 | - | 0.912 ± 0.056 | 0.948 ± 0.001 | 0.952 ± 0.002 | - |
| | Fish | *0.883 ± 0.010* | 0.144 ± 0.158 | 0.072 ± 0.054 | - | 0.922 ± 0.006 | 0.881 ± 0.006 | 0.878 ± 0.008 | - |
| | Mixup | 0.288 ± 0.498 | 0.129 ± 0.217 | 0.126 ± 0.218 | - | *0.940 ± 0.011* | **0.949 ± 0.003** | 0.947 ± 0.006 | - |
| | MMD | 0.835 ± 0.047 | 0.809 ± 0.027 | 0.766 ± 0.030 | - | 0.922 ± 0.027 | 0.946 ± 0.004 | 0.947 ± 0.006 | - |
| | DeepCoral | 0.852 ± 0.036 | **0.813 ± 0.017** | **0.782 ± 0.022** | - | *0.940 ± 0.007* | 0.948 ± 0.004 | 0.949 ± 0.003 | - |
| | GroupDRO | 0.869 ± 0.034 | *0.811 ± 0.028* | 0.761 ± 0.056 | - | 0.919 ± 0.027 | 0.947 ± 0.006 | **0.953 ± 0.001** | - |
| FL | FedProx | - | 0.126 ± 0.237 | 0.121 ± 0.245 | 0.000 ± 0.000 | - | 0.939 ± 0.002 | 0.935 ± 0.001 | *0.937 ± 0.007* |
| | Scaffold | - | 0.728 ± 0.010 | *0.776 ± 0.016* | **0.800 ± 0.017** | - | 0.942 ± 0.001 | 0.929 ± 0.003 | 0.932 ± 0.001 |
| | AFL | - | 0.811 ± 0.024 | 0.834 ± 0.005 | 0.828 ± 0.005 | - | 0.942 ± 0.011 | 0.947 ± 0.006 | 0.932 ± 0.008 |
| FDG | FedDG | - | 0.561 ± 0.184 | 0.531 ± 0.112 | 0.464 ± 0.201 | - | 0.863 ± 0.012 | 0.856 ± 0.003 | 0.869 ± 0.016 |
| | FedADG | - | 0.674 ± 0.003 | 0.669 ± 0.008 | *0.600 ± 0.009* | - | 0.936 ± 0.001 | 0.933 ± 0.004 | 0.926 ± 0.002 |
| | FedSR | - | 0.381 ± 0.018 | 0.363 ± 0.041 | 0.383 ± 0.015 | - | 0.934 ± 0.010 | 0.917 ± 0.009 | 0.925 ± 0.003 |
| | FedGMA | - | 0.620 ± 0.012 | 0.615 ± 0.018 | 0.487 ± 0.008 | - | 0.901 ± 0.006 | 0.854 ± 0.006 | 0.815 ± 0.025 |

Table 10: Test accuracy on CivilComments and Py150 datasets with held-out-domain validation where FedAvg-ERM is the simple baseline (B). "-" means the method is not applicable in that context. Bold is for best and italics is for second best in each column. FedDG and FedADG are designed for image dataset and thus not applicable for CivilComments and Py150. We report the standard deviation among 3 runs.

| | | CivilComments ($D=16$) | | | | Py150 ($D=5477$) | | | |
|---|---|---|---|---|---|---|---|---|---|
| | | $C=1$ | $C=100$ (FL) | | | $C=1$ | $C=100$ (FL) | | |
| | | (centralized) | $\lambda=1$ | $\lambda=0.1$ | $\lambda=0$ | (centralized) | $\lambda=1$ | $\lambda=0.1$ | $\lambda=0$ |
| B | FedAvg-ERM | 0.541 ± 0.003 | 0.359 ± 0.028 | 0.347 ± 0.018 | 0.325 ± 0.009 | **0.683 ± 0.003** | **0.683 ± 0.000** | *0.650 ± 0.000* | *0.641 ± 0.000* |
| DG Adapted | IRM | 0.641 ± 0.001 | *0.587 ± 0.018* | 0.534 ± 0.038 | - | *0.677 ± 0.000* | *0.655 ± 0.001* | **0.653 ± 0.000** | 0.627 ± 0.001 |
| | Fish | **0.671 ± 0.000** | 0.424 ± 0.158 | 0.340 ± 0.173 | - | 0.663 ± 0.000 | 0.650 ± 0.000 | 0.648 ± 0.001 | **0.642 ± 0.001** |
| | MMD | *0.652 ± 0.000* | **0.634 ± 0.012** | **0.613 ± 0.014** | - | 0.656 ± 0.001 | 0.627 ± 0.000 | 0.629 ± 0.000 | 0.610 ± 0.000 |
| | DeepCoral | 0.585 ± 0.000 | 0.515 ± 0.061 | 0.463 ± 0.080 | - | 0.656 ± 0.003 | 0.651 ± 0.000 | 0.650 ± 0.001 | 0.639 ± 0.000 |
| | GroupDRO | 0.638 ± 0.003 | 0.475 ± 0.014 | *0.474 ± 0.003* | - | 0.513 ± 0.001 | 0.589 ± 0.000 | 0.603 ± 0.003 | 0.607 ± 0.002 |
| FL | FedProx | - | 0.181 ± 0.032 | 0.173 ± 0.033 | 0.170 ± 0.036 | - | 0.638 ± 0.000 | 0.633 ± 0.000 | 0.612 ± 0.000 |
| | Scaffold | - | 0.389 ± 0.015 | 0.376 ± 0.018 | *0.332 ± 0.014* | - | 0.642 ± 0.000 | 0.638 ± 0.000 | 0.617 ± 0.000 |
| | AFL | - | 0.552 ± 0.021 | 0.474 ± 0.014 | **0.435 ± 0.019** | - | 0.486 ± 0.000 | 0.485 ± 0.003 | 0.467 ± 0.001 |
| FDG | FedSR | - | 0.360 ± 0.003 | 0.339 ± 0.003 | 0.319 ± 0.003 | - | 0.533 ± 0.002 | 0.526 ± 0.001 | 0.445 ± 0.003 |
| | FedGMA | - | 0.209 ± 0.028 | 0.202 ± 0.024 | 0.200 ± 0.022 | - | 0.620 ± 0.000 | 0.613 ± 0.000 | 0.600 ± 0.000 |

## H TRAINING TIME, COMMUNICATION ROUNDS AND LOCAL COMPUTATION

In this section, we provide training time per communication in terms of the wall clock training time. Notice that for a fixed dataset, most of algorithms have similar training time comparing to FedAvg-ERM, where FedDG and FedADG are significantly more expensive.

Table 11: Test accuracy on FEMNIST dataset with held-out validation where FedAvg-ERM is the simple baseline (B). "-" means the method is not applicable in that context. Bold is for best and italics is for second best in each column.

| | | FEMNIST (Held-Out-Domain) | | | |
|---|---|---|---|---|---|
| | | $C = 1$ | $C = 100$ (FL) | | |
| | | (centralized) | $\lambda = 1$ | $\lambda = 0.1$ | $\lambda = 0$ |
| B | FedAvg-ERM | $\mathbf{0.854 \pm 0.001}$ | $0.837 \pm 0.003$ | $0.838 \pm 0.002$ | $0.834 \pm 0.001$ |
| DG Adapted | IRM | $0.844 \pm 0.001$ | $0.832 \pm 0.007$ | $0.822 \pm 0.005$ | $0.821 \pm 0.004$ |
| | Fish | $0.849 \pm 0.002$ | $0.833 \pm 0.004$ | $0.829 \pm 0.003$ | $0.826 \pm 0.008$ |
| | Mixup | $0.834 \pm 0.002$ | $0.828 \pm 0.009$ | $0.821 \pm 0.008$ | $0.816 \pm 0.005$ |
| | MMD | $0.844 \pm 0.003$ | $0.843 \pm 0.006$ | $0.832 \pm 0.007$ | $0.829 \pm 0.006$ |
| | DeepCoral | $0.846 \pm 0.002$ | $0.840 \pm 0.005$ | $0.836 \pm 0.002$ | $0.851 \pm 0.003$ |
| | GroupDRO | $0.841 \pm 0.004$ | $0.835 \pm 0.005$ | $0.814 \pm 0.002$ | $0.805 \pm 0.003$ |
| FL | FedProx | - | $\mathbf{0.865 \pm 0.001}$ | $\mathbf{0.864 \pm 0.001}$ | $\mathbf{0.863 \pm 0.001}$ |
| | Scaffold | - | $0.860 \pm 0.002$ | $0.859 \pm 0.002$ | $0.858 \pm 0.002$ |
| | AFL | - | $0.629 \pm 0.010$ | $0.593 \pm 0.037$ | $0.584 \pm 0.020$ |
| FDG | FedDG | - | $0.859 \pm 0.002$ | $0.857 \pm 0.002$ | $0.856 \pm 0.002$ |
| | FedADG | - | $0.817 \pm 0.026$ | $0.800 \pm 0.061$ | $0.785 \pm 0.025$ |
| | FedSR | - | $0.832 \pm 0.015$ | $0.832 \pm 0.020$ | $0.831 \pm 0.014$ |
| | FedGMA | - | $0.849 \pm 0.010$ | $0.842 \pm 0.004$ | $0.837 \pm 0.003$ |

Table 12: Gap Table: Current Progress in solving DG in FL context

| | | Domain Seperation | Free of Data Leakage | PACS | IWildCam | CelebA | Camelyon17 | CivilComments | Py150 |
|---|---|---|---|---|---|---|---|---|---|
| | Centralized adopted methods | ✗ | ✓ | −0.010 | −0.047 | +0.265 | +0.003 | +0.266 | +0.003 |
| | Federated methods | ✓ | ✓ | −0.040 | +0.010 | +0.319 | −0.003 | +0.266 | −0.012 |
| FDG | FedDG | ✓ | ✗ | −0.008 | −0.010 | +0.014 | −0.094 | - | - |
| | FedADG | ✓ | ✓ | −0.013 | +0.060 | +0.152 | −0.017 | - | - |
| | FedSR | ✓ | ✓ | −0.407 | −0.104 | −0.154 | −0.033 | −0.008 | −0.124 |
| | FedGMA | ✓ | ✓ | −0.225 | −0.094 | +0.098 | −0.096 | −0.145 | −0.037 |

Table 13: Wall-clock Training time per communication (unit: s). There might be variance due to the machine's status and resource availability.

| | | PACS $C = 100$ | IWildCam $C = 243$ | CelebA $C = 100$ | Camelyon17 $C = 100$ | CivilComments $C = 100$ | Py150 $C = 100$ | FEMNIST $C = 100$ |
|---|---|---|---|---|---|---|---|---|
| B | FedAvg-ERM | 143 | 6301 | 298 | 845 | 3958 | 6566 | 262 |
| DG Adapted | IRM | 147 | 6454 | 309 | 1163 | 4085 | 7089 | 297 |
| | Fish | 148 | 7072 | 312 | 1003 | 5483 | 7770 | 324 |
| | Mixup | 144 | 6294 | 302 | 933 | - | - | 264 |
| | MMD | 144 | 6663 | 312 | 963 | 4024 | 7603 | 287 |
| | DeepCoral | 144 | 6597 | 313 | 879 | 3901 | 7212 | 287 |
| | GroupDRO | 145 | 9311 | 310 | 750 | 4690 | 8121 | 307 |
| FL | FedProx | 169 | 6921 | 1219 | 4310 | 4502 | 6513 | 288 |
| | Scaffold | 167 | 6876 | 1344 | 4623 | 4421 | 6389 | 281 |
| FDG | FedDG | 352 | 32172 | 9232 | 15784 | - | - | 989 |
| | FedADG | 181 | 11094 | 6502 | 9945 | - | - | 907 |
| | FedSR | 151 | 7136 | 2267 | 6567 | 4403 | 8020 | 280 |
| | FedGMA | 143 | 6795 | 558 | 2036 | 4525 | 6545 | 261 |

