# OpenReview forum: "Benchmarking Algorithms for Federated Domain Generalization"
_ICLR.cc/2024/Conference — ICLR 2024 spotlight_

### Official Review · Reviewer_EtKR · 2023-10-24

**Soundness:** 4 excellent
**Presentation:** 4 excellent
**Contribution:** 3 good
**Rating:** 8
**Confidence:** 3

**Summary:**

This paper explores the intersection between Federated Learning and Domain Generalization, namely, Federated Domain Generalization (FDG), in which different domains are separated among clients that will collaboratively learn a model that generalizes to unseen domains. This paper pushes on an important direction: on the methodology behind evaluating FDG algorithms. In this sense, the authors: (i) Present an interesting review of existing practice in FDG; (ii) Propose a novel way of partitioning clients in FDG; (iii) Propose new metrics for evaluating the hardness of benchmarks; (iv) provide extensive evaluation of FDG methods.

**Strengths:**

This paper plays the same role of [Gulrajani and Lopez-Paz, 2021] for FDG, i.e., an important paper that provides an in-depth discussion about how to evaluate existing methods. In this angle, the paper provides extensive experimentation and interesting insights. In this sense, the paper is quite important for the field.

Furthermore, as the authors discuss in the paper, the federated setting poses __new challenges__ to DG. This is especially related to the new partition method that the authors propose, and the main novelty of the paper. This contribution helps merging the fields of federated learning and domain generalization, making the evaluation of FDG algorithms more realistic. As a consequence, this direction is quite impactful and helpful for the field of FDG.

**Weaknesses:**

Overall, I have no major concerns with this paper. My only critique is a (minor) lack of clarity. In section 4., the term ERM is never defined, and the empirical risk minimization method is never properly described. While, for knowledgeable audiences, this does not hurt the understanding of the paper in general, this makes the paper harder to read for beginners. I suggest authors add a description of the ERM in page 3, on the Federated DG paragraph.

**Questions:**

__Q1.__ Since Domain Adaptation is a related problem to Domain Generalization, is it possible to apply the proposed partitioning scheme to Federated Domain Adaptation?

---

> ### Author Response · Authors · 2023-11-19
> **Response to Reviewer EtKR (1/1)**
>
> We appreciate the reviewer's careful reading and recognition of this work. Especially for the suggestions on expanding the data partition method on other scenarios. We next provide a detailed reply as below.
>
> > My only critique is a (minor) lack of clarity. In section 4., the term ERM is never defined, and the empirical risk minimization method is never properly described. While, for knowledgeable audiences, this does not hurt the understanding of the paper in general, this makes the paper harder to read for beginners. I suggest authors add a description of the ERM in page 3, on the Federated DG paragraph.
>
> Thanks for pointing this out! We have revised manuscript on page 3:
> “The most common objective in Federated learning is empirical risk minimization (ERM), which minimizes the average loss over the given dataset.”
>
> > Since Domain Adaptation is a related problem to Domain Generalization, is it possible to apply the proposed partitioning scheme to Federated Domain Adaptation?
>
> Thank you for your insightful question. Yes, our partition method is general for partitioning of any D distributions onto C clients; the distributions could be defined based on:
> - Domain labels as in DG in our paper or domain adaptation as reviewer mentioned.
> - Class labels as in class-heterogeneous FL.
> - Sensitive attributes in FL fairness [1,2].
>
> We have added the discussion in the main paper at the start of section 3 heterogeneous partitioning method.
>
> Modification on page 3: “Generally, our partition method effectively handles partitioning $D$ types of integer-numbered objects into $C$ groups. It's broadly applicable, suitable for domain adaptation, ensuring fairness in Federated Learning (FL), and managing non-iid FL regimes. ”
>
> [1] Mohri, Mehryar, Gary Sivek, and Ananda Theertha Suresh. "Agnostic federated learning." International Conference on Machine Learning. PMLR, 2019.
>
> [2] Du, Wei, et al. "Fairness-aware agnostic federated learning." Proceedings of the 2021 SIAM International Conference on Data Mining (SDM). Society for Industrial and Applied Mathematics, 2021.

---

> > ### Comment · Reviewer_EtKR · 2023-11-20
> > **Response to Authors**
> >
> > Dear authors,
> >
> > Thank you for your response, and for integrating my suggestions into your paper. In my view this paper can have an important impact on the field of federated domain generalization.
> >
> > As I stated in my initial review, I had no major concens with this paper. As a result, I keep my score __8: Accept, Good Paper__.

---

### Official Review · Reviewer_AnGM · 2023-11-01

**Soundness:** 3 good
**Presentation:** 3 good
**Contribution:** 3 good
**Rating:** 6
**Confidence:** 3

**Summary:**

This paper proposes a benchmark for domain generalization (DG) in federated learning. Specifically, they (1) develop a novel method to partition a DG dataset to any number of clients, (2) propose a benchmark methodology including four important factors, (3) experiment with a line of baselines and datasets.

**Strengths:**

1. This paper contributes to an important topic. I believe federated DG is an important problem in FL, especially cross-device FL when the trained FL global model need to generalize to a large amount of clients that do not participate in FL training.
2. The paper is well-written and easy to follow.
3. The proposed benchmark includes a variety of datasets and algorithms.
4. The experiments regarding the number of clients is highly related to cross-device FL, and indicate an important drawback in the current federated DG experiments.

**Weaknesses:**

1. In the context of domain generalization, we are particularly concerned with whether models trained on limited source domains can generalize to new target domains. This paper also uses held-out domains (in Appendix C.2). I believe that this aspect should be more explicitly explained in the main body of the text; otherwise, readers might easily misconstrue that all domains were used to construct training clients, which is misleading.
2. Error bars are not provided for experiment results, the conclusion may be influence by random fluctuation.

Minor:
1. Page5 line 12: homogeneous -> homogeneity
2. When considering the second kind of DG, it is relevant to “performance fairness” in FL, which encourage a uniform distribution of accuracy across clients. Although works in this direction might emphasize more on participating clients, I believe at least the AFL algorithm [1] can be a good supplement to the benchmark.

[1] Mehryar Mohri, Gary Sivek, Ananda Theertha Suresh. Agnostic Federated Learning. ICML 2019.

**Questions:**

1. In Eq. (1), two kinds of DG is mentioned. Which kind of DG is mainly used in your benchmark?
2. The performance for some algorithm, for example, FedSR, is very low, and consistently lower than FedAvg. Could you explain the reason behind?

---

> ### Author Response · Authors · 2023-11-19
> **Response to Reviewer AnGM (1/2)**
>
> We appreciate the reviewer for the positive feedback and construcive suggestions. We responded to the questions and suggestions you made individually below.
>
> > In the context of domain generalization, we are particularly concerned with whether models trained on limited source domains can generalize to new target domains. This paper also uses held-out domains (in Appendix C.2). I believe that this aspect should be more explicitly explained in the main body of the text...
>
> Good point. We agree that our setup of held-out validation domains is of great importance and need better visibility. We have added the corresponding part to the main paper section 4.3, page 7 per the reviewer’s suggestion, where we emphasize that we are using validation domain dataset: “In DG task, we cannot access the test domain data. However, we are particularly concerned about the model performance outside the training domains, thus we preserve a small portion of the domains we can access as held-out validation domains, and the held-out validation domains are used for model selection and early stopping. ”
>
>
> > Error bars are not provided for experiment results, the conclusion may be influenced by random fluctuation.
>
> Great suggestion. Following the Reviewer's suggestion, we have included in the revised manuscript error bar in Section 4.3. In particular, we have included the standard deviation of the repeated experiments on PACS, CelebA, Cameylon17. Due to the computational challenges of training many different models with different settings, we keep working on the results on Py150, CivilComments, IWildCam and FEMNIST datasets, and we will include them in the final version.
>
> > Page5 line 12: homogeneous -> homogeneity
>
> Thanks, fixed.
>
> > When considering the second kind of DG, it is relevant to “performance fairness” in FL, which encourages a uniform distribution of accuracy across clients. Although works in this direction might emphasize more on participating clients, I believe at least the AFL algorithm [1] can be a good supplement to the benchmark.
>
> We appreciate the reviewer's mention of this work. In a nutshell, the Reviewer is right:  Agnostic Federated Learning (AFL) shares similarities with Domain Generalization in a Federated context. This is evident as both approaches address scenarios where the test distribution diverges from the training distribution. Thus we agree that AFL is a good method to evaluate especially when tackling subpopulation shift tasks. We have added a discussion in the revised paper page 17.  Furthermore, we are currently working on implementing AFL in our benchmark, especially for CelebA, CivilComments, which are designed for the sub-population shift task. Our benchmark could also help the evaluation of the future work in this line of research.
>
> Modification on paper page 17: “Agnostic Federated Learning (AFL) [1,2] share similarities with Domain Generalization in a Federated context. This is evident as both approaches address scenarios where the test distribution diverges from the training distribution. AFL constructs a framework that naturally yields a notion of fairness, where the centralized model is optimized for any target distribution formed by a mixture of the client distributions. Thus, AFL is a good method to evaluate especially when tackling subpopulation shift tasks.”
>
> > In Eq. (1), two kinds of DG are mentioned. Which kind of DG is mainly used in your benchmark?
>
> Great question. We used both formulations. We elaborate this both in terms of methods and datasets.
>
> Regarding methods, since test domains are inaccessible for direct optimization, DG methods typically start from one of the two theoretical formulations and seek to find an approximate objective on training domains to work with. For instance, Fish starts with the average-case objective, and IRM, GroupDRO starts with the worst-case objective. We included diverse methods from both formulations.
>
> Regarding datasets, PACS, FEMNIST, Camelyon17, and IWildCam focus on averaging test domain performance, aligning with the first DG objective for evaluation. Among them, IWildCam uses the F1 Score, which is a weighted average, placing greater emphasis on rarer species. In contrast, CelebA and CivilComments prioritize worst domain accuracy, aligning with the second DG objective. Further, Py150 is unique in targeting a specific sub-population accuracy, termed 'method-class accuracy.' This is particularly relevant for Py150's application in code completion, where the accuracy of predicting method and class names is critical.
>
> [1] Mohri, Mehryar, Gary Sivek, and Ananda Theertha Suresh. "Agnostic federated learning." International Conference on Machine Learning. PMLR, 2019.
>
> [2] Du, Wei, et al. "Fairness-aware agnostic federated learning." Proceedings of the 2021 SIAM International Conference on Data Mining (SDM). Society for Industrial and Applied Mathematics, 2021.

---

> ### Author Response · Authors · 2023-11-19
> **Response to Reviewer AnGM (2/2)**
>
> > The performance for some algorithms, for example, FedSR, is very low, and consistently lower than FedAvg. Could you explain the reason behind?
>
> Thank you for your question. First, we would like to point out that we replicated the results in the experimental setting of the FedSR paper, where their evaluation on the PACS dataset is only based on 3 clients. In our benchmark, we observed two new phenomena for FedSR:
> - the degradation in performance when the number of clients is large (increase up to $100$)
> - the hyperparameters are also more sensitive compared to their 3 clients setting.
>
> As suggested by reviewer AnGM, we conducted repeated experiments (Sec. 4.3), where we provided each method with a larger hyperparameter tuning budget, thus, FedSR achieves better results than in the original experiments. Even with an increased budget, we observed that FedSR is still significantly worse than FedAvg ($40\\%$ vs $90\\%$ on PACS). Thus, it does not alter the conclusion of the original manuscript.
>
> **A potential explanation of the phenomena**
>
> The convergence analysis is well established for FedAvg in the literature, see [3] and subsequent works. However, there’s no theoretical guarantees for FedSR, where the objective function is constructed by the summation of local data loss and local non-smooth penalty. We suspect that the extra heterogeneous local nonsmooth penalty terms introduced in FedSR potentially diverts each client's learning trajectory, leading to harder convergence when client number increases.
>
> [3] Stich, Sebastian U. "Local SGD converges fast and communicates little." arXiv preprint arXiv:1805.09767 (2018).

---

> > ### Comment · Reviewer_AnGM · 2023-11-22
> > **Thanks!**
> >
> > Thanks a lot for your rebuttal! I think the additional information you provide is beneficial to the paper. I understand that providing error bars for all experiments can be challenging in such a short period of time during rebuttal, but I do believe that a good and tested benchmark is beneficial for the whole community. Good luck!

---

> > > ### Author Response · Authors · 2023-11-23
> > > **Further Response to Reviewer AnGM**
> > >
> > > Thank you! We express our sincere appreciation to the reviewer for the valuable suggestions, particularly for highlighting the significance of AFL. Here, we report the detailed discussion of AFL, and our modification of AFL in the Benchmark, as well as the initial results on CelebA.
> > >
> > > First, we would like to point out that AFL is designed for client number $C$ matches domain number $D$, where each client is assumed to be a domain. Furthermore, it requires communication at every iteration. In this benchmark, we have made the following modifications to AFL to accommodate the general $C \neq D$ cases and to reduce communications:
> > > - To allow $C\neq D,$ we construct new objective as the following:
> > > $$\min\_\theta\max\_{\beta\in\Delta\_D} \mathcal{L}(\theta, \beta) = \sum\_{c=1}^{C}\mathcal{L}\_c(\theta,\beta)=\sum\_{c=1}^{C}\sum\_{d=1}^{D}\beta\_d\ell\_{d,c}(\theta).$$
> > > - To reduce communication cost, we allow $\theta$ to be updated multiple iterations locally per communication. Further, we maintain the update of $\beta$ on the central server, as in AFL, given that it requires projection of the global updates onto the simplex. This projection is not equivalent to averaging the locally projected updates of  $\beta.$
> > >
> > > Please see Algorithm 2 for pseudo code in the revised manuscript page 17.
> > >
> > > Then, we would like to report the result of modified AFL on CelebA dataset. It currently achieves very high accuracy. We are currently running it multiple times to get the error bar, as well as running on other datasets.
> > >
> > > |              | Centralized | $\lambda=1$ | $\lambda=0.1$ | $\lambda=0$ |
> > > |--------------|-------------|-------------|---------------|-------------|
> > > | Modified AFL | $0.8667$    | $0.8556$    | $0.8512$      | $0.8435$    |

---

### Official Review · Reviewer_PJ3y · 2023-11-26

**Soundness:** 3 good
**Presentation:** 3 good
**Contribution:** 3 good
**Rating:** 6
**Confidence:** 2

**Summary:**

This paper introduces a benchmark for federated domain generalization, which is a challenging problem that requires learning a model that can generalize to heterogeneous data in a federated setting. The paper presents a novel data partitioning method that can create heterogeneous clients from any domain dataset, and a benchmark methodology that considers four factors: number of clients, client heterogeneity, dataset diversity, and out-of-distribution generalization. The paper also evaluates 13 Federated DG methods on 7 datasets and provides insights into their strengths and limitations.

**Strengths:**

* The paper proposes a comprehensive and rigorous benchmark for Federated DG that covers various aspects of the problem and can be easily extended to new datasets and methods.
* The paper provides a clear and detailed description of the data partitioning method and the benchmark methodology, as well as the implementation details of the methods and datasets.
* The paper conducts extensive experiments and analyzes the results from different perspectives.

**Weaknesses:**

The paper mainly summarizes the experimental observations, but does not offer much theoretical analysis or explanation for why some methods perform better than others in certain scenarios, which would be helpful to gain more insights into the FDG problem and the design of effective methods.

**Questions:**

No.

---

> ### Author Response · Authors · 2023-11-30
> **Response to Reviewer PJ3y (1/1)**
>
> > The paper mainly summarizes the experimental observations, but does not offer much theoretical analysis or explanation for why some methods perform better than others in certain scenarios, which would be helpful to gain more insights into the FDG problem and the design of effective methods.
>
> We appreciate the reviewer for the comments and feedback. While we cannot fully explain all performance differences, we suggest a few possible explanations and general insights that may help in designing more effective federated DG methods.
>
> **Mismatch between method’s (implicit) assumptions vs real-world datasets** One potential issue is that each method may implicitly or explicitly assume certain conditions about the data generating process. For instance, IRM assumes the special structural equation model and FedDG assumes that the amplitude spectrum mainly consists of style information and the phase spectrum mainly consists of object information. Yet, the underlying data generating process may not match these assumptions. This mismatch may lead to varying relative performance across datasets. More careful analysis of the implicit or explicit assumptions of each method compared to real-world datasets could be a fruitful future research direction.
>
> **Privacy Constraint** Centralized domain generalization methods usually assume that the data from each domain are accessible. Yet, in practice, each client may only have access to one domain (our domain separation case). Thus, centralized methods are not directly applicable in the domain separation case. A potential research direction for this could be designing a DG objective to be decomposable across domains so that it is amenable for FL.
>
> **Convergence in FL setting** While in a centralized setting, it is often assumed that the method will naturally converge using an appropriate learning rate, the speed of convergence and solution at convergence may have a much higher impact on DG in the FL setting. For example, FedSR and FedGMA may generalize well but the convergence is very slow when the number of clients is large.  On the other hand, the convergence analyses are well established for FedAvg, FedProx, Scaffold in the literature, see [2] and subsequent works. However, there’s no theoretical guarantees for newly designed methods for Federated domain generalization. Because FedSR’s objective function is constructed by the summation of local data loss and local non-smooth penalty, we suspect that the heterogeneous local nonsmooth penalty terms introduced in FedSR potentially diverts each client's learning trajectory, leading to slow convergence when client number increases. For other methods that are designed to converge quickly, like FedProx and Scaffold, the solution they converge to does not generalize well. This is because they were designed to address client heterogeneity but were not designed for train-test heterogeneity (i.e., distribution shifts). Therefore, it is unsurprising that they do not perform well on DG tasks. Given all of this, a potential research direction would be to combine the strengths of provably convergent FL algorithms with DG objectives or regularizations. It seems that for good federated DG performance, both aspects need to be solved simultaneously.
> Thus, how to design a method generalize better than simple FedAVG and maintaining the convergence property of FedAvg is of great importance. This is the message we wish to convey in the benchmark.  We will incorporate those explanations in the paper to provide intuition on solving domain generalization in the Federated context.
>
> [1] Koh, Pang Wei, et al. "Wilds: A benchmark of in-the-wild distribution shifts." International Conference on Machine Learning. PMLR, 2021.
>
> [2] Stich, Sebastian U. "Local SGD converges fast and communicates little." arXiv preprint arXiv:1805.09767 (2018).

---

### Author Response · Authors · 2023-11-19
**General Reply**

We sincerely appreciate the reviewers for their insightful and helpful comments. We also thank the AC for handling our submission. We have revised the paper based on the excellent feedback, as detailed below; all the changes in the revised manuscript have been reported in blue color.

---

### Meta-Review · Area_Chair_tkqP · 2023-12-20

**Metareview:**

The reviewers unanimously recommended accept, while hoping some remaining issues could be fixed in the next version (e.g., Error bars). Overall, reviewers believe have a well-designed benchmark for Federated Domain Generalization is timely and important, and this paper provides a great framework for that.

**Justification For Why Not Higher Score:**

N/A

**Justification For Why Not Lower Score:**

This paper provides a solid solution to a timely problem

---

### Decision · Program_Chairs · 2024-01-16

Accept (spotlight)